# Effect of crystal facets in plasmonic catalysis

Yicui Kang [1,8], Simão M. João [2,8], Rui Lin [1] ✉, Kang Liu[3], Li Zhu[1], Junwei Fu[3], Weng-Chon (Max) Cheong[4], Seunghoon Lee [1,5,6], Kilian Frank [7], Bert Nickel [7], Min Liu [3], Johannes Lischner[2] ✉ & Emiliano Cortés [1] ✉

While the role of crystal facets is well known in traditional heterogeneous catalysis, this effect has not yet been thoroughly studied in plasmon-assisted catalysis, where attention has primarily focused on plasmon-derived mechanisms. Here, we investigate plasmon-assisted electrocatalytic $CO_2$ reduction using different shapes of plasmonic Au nanoparticles - nanocube (NC), rhombic dodecahedron (RD), and octahedron (OC) - exposing {100}, {110}, and {111} facets, respectively. Upon plasmon excitation, Au OCs doubled CO Faradaic efficiency ($FE_{CO}$) and tripled CO partial current density ($j_{CO}$) compared to a dark condition, with NCs also improving under illumination. In contrast, Au RDs maintained consistent performance irrespective of light exposure, suggesting minimal influence of light on the reaction. Temperature experiments ruled out heat as the main factor to explain such differences. Atomistic simulations and electromagnetic modeling revealed higher hot carrier abundance and electric field enhancement on Au OCs and NCs than RDs. These effects now dominate the reaction landscape over the crystal facets, thus shifting the reaction sites when comparing dark and plasmon-activated processes. Plasmon-assisted $H_2$ evolution reaction experiments also support these findings. The dominance of low-coordinated sites over facets in plasmonic catalysis suggests key insights for designing efficient photocatalysts for energy conversion and carbon neutralization.

Fossil fuels have been the dominant source of cheap energy for the past decades, but excessive use has caused the release of large amounts of $CO_2$ into the atmosphere, producing harmful consequences in terms of climate change and energy crises. While a complete solution to this problem will also involve investing in more sustainable energy generation alternatives, decarbonization is a critical step in its path. Artificial photosynthesis offers an attractive implementation to these solutions by mimicking the natural process of reducing $CO_2$ into chemical fuels using sunlight[1]. However, the $CO_2$ reduction reaction ($CO_2RR$) is a multi-electron and multi-proton process, leading to many reaction pathways. Therefore, achieving high selectivity has always been a challenge for $CO_2RR$[2-4]. As a result, efforts to improve the performance of catalysts have been widely made, including modifying the component[4-9], size[5,10], geometric structure[11-14], as well as the facets[15-17] of the materials. Among them, the crystal facet effect is regarded as a crucial factor that can affect catalytic activity and selectivity by tuning, for example, the atomic arrangement[18] or the adsorption energy of intermediates[19,20], among

[1]Nanoinstitute Munich, Faculty of Physics, Ludwig-Maximilians-Universität München, 80539 München, Germany. [2]Departments of Materials and Physics and the Thomas Young Centre for Theory and Simulation of Materials, Imperial College London, London, UK. [3]Hunan Joint International Research Center for Carbon Dioxide Resource Utilization, School of Physics and Electronics, Central South University, 410083 Changsha, China. [4]Macao Institute of Materials Science and Engineering (MIMSE), Faculty of Innovation Engineering (FIE), University of Science and Technology, Macau SAR 999078, P. R. China. [5]Department of Chemistry, Dong-A University, Busan 49315, South Korea. [6]Department of Chemical Engineering (BK21 FOUR Graduate Program), Dong-A University, Busan 49315, South Korea. [7]Faculty of Physics and Center for Nanoscience (CeNS), Ludwig-Maximilians-Universität, Geschwister-Scholl-Platz 1, 80539 München, Germany. [8]These authors contributed equally: Yicui Kang, Simão M. João. ✉e-mail: Rui.Lin@physik.uni-muenchen.de; j.lischner@imperial.ac.uk; Emiliano.Cortes@lmu.de

others. Besides, distinctive facets also exhibit different physical properties, such as work function[21], electronic states[22], and electron mean free path[23]. Recent years have witnessed considerable research into the facet effect on heterogeneous catalysts, including thermocatalysis[24–26], electrocatalysis[18,19,27–37], photocatalysis[38,39] and photo-electrocatalysis[40,41]. As an illustration, Hori et al. investigated the facet effect of single crystal Pt in electrocatalytic $CO_2RR$ systems dating back to 1995. Their study demonstrated that Pt (110) displayed a CO formation rate exceeding ten times that of Pt (111), highlighting the substantial promise of tailoring exposed facets to modulate the selectivity and activity of catalysts for $CO_2RR$[37].

In parallel, the emergence of plasmonic catalysis as a novel subfield in heterogeneous catalysis has garnered significant attention due to its distinct properties and potential for enhancing activity and selectivity in diverse catalytic processes. Plasmonic catalysis involves the light excitation of localized surface plasmons (LSP), which refers to the resonant oscillation of conduction electrons induced by photons[6]. At the LSP resonance (LSPR), nanoparticles (NPs) concentrate electromagnetic fields on their surfaces, leading to a significant field enhancement. Subsequently, the surface plasmons decay into energetic electron and hole pairs through Landau damping. These carriers with high energies can then be transferred into unoccupied levels of acceptor molecules adsorbed on the catalyst surface and induce chemical transformations or selective desorption, among others[42,43]. Furthermore, plasmonic catalysts may have the capability to selectively activate chemical bonds of adsorbed molecules, thereby opening avenues for targeting specific chemical pathways within a reaction[43–45]. On top of these electronic effects, the temperature increase associated with the plasmon decay can further contribute to the chemical reaction[46,47]. As such, many reactions have been studied under this framework, such as $H_2$ dissociation[48,49], $O_2$ activation[50], $N \equiv N$ dissociation[51], $H_2$ evolution[52,53], as well as $CO_2RR$[54]. These studies demonstrated the ability of plasmonic catalysis to enhance the efficiency of chemical reactions or even modify reaction pathways[7,55,56]. Moreover, the LSP offers high tunability and presents opportunities to manipulate light absorption at nanometer and femtosecond scales[57]. These remarkable features position plasmonic catalysis as a promising avenue for enhancing the efficiency and selectivity of solar energy conversion[58]. Yet the rational design of plasmonic catalysts is still in its infancy. Hence, the primary objective here is to study the effect of crystal facets in plasmonic $CO_2RR$, building on previous evidence of its significance in thermal catalysis, electrocatalysis, photocatalysis, and related areas.

Drawing from the aforementioned fundamentals, we synthesized 3 morphologies of Au nanoparticles (NPs)—nanocubes (NCs), rhombic dodecahedrons (RDs), and octahedrons (OCs)—with the same phase, size and LSPR position, but distinctive exposed facets. The electrocatalytic $CO_2RR$ response on the three different types of Au NPs differ significantly in between them, given the different $CO_2$ activation energy on each of the exposed crystal facets, as confirmed by the DFT calculations. However, when introducing plasmons into the system, two of them—Au OCs and NCs—exhibit significant increase in the selectivity to CO, while RDs do not show any improvement. Large-scale atomistic simulations and electric field modeling were conducted to investigate the underlying mechanism. Our findings reveal that the response to plasmons in catalytic processes predominantly relies on the amount and spatial distribution of plasmon-induced hot carriers and electric field enhancement. In this context, edges and corners are pivotal while the role of facets appears to be insignificant. These results offer insights for potentially unlocking pathways to harness sunlight directly for decarbonization processes.

## Results

### Synthesis and characterization of Au NPs

Gold nanocrystals with three different exposed facets—{100}, {110}, and {111}—were prepared by seed-mediated growth approaches in the presence of CTAB surfactant[59,60]. The different exposed facets resulted in different morphologies, such as nanocubes (NCs), rhombic dodecahedra (RDs), and octahedra (OCs), respectively. The detailed synthetic process for each catalyst is described in the Methods section. SEM and TEM images of the synthesized single-crystalline Au nanocrystals with monodispersed shapes and sizes are shown in Fig. 1a–c and Fig. S1. The Au NCs, RDs, and OCs have average sizes of around 60 nm (Fig. S2). The face-centered cubic (FCC) phase of all three Au nanoparticles was confirmed by X-ray diffraction (XRD) (Fig. 1g). To gain further insights into the surface structure of the Au nanoparticles, high-resolution transmission electron microscopy (HRTEM) and selected area electron diffraction (SAED) were performed, as depicted in Fig. 1d–f. The observed d-spacing values in HRTEM images of Au NCs, RDs, and OCs were 0.209, 0.141, and 0.242 nm, respectively. These values align with the surface termination of Au {100}, {110}, and {111} facets in the face-centered cubic (FCC) phase, as presented in Table S1. The distinctly sharp and well-defined spots in SAED patterns confirmed the single-crystalline structure of Au NPs. These findings are also consistent with existing literature on crystal facet characterization of Au NPs[59]. UV−vis−NIR absorption spectroscopy was employed to examine the optical properties of Au NPs. Figure 1h illustrates that all three samples exhibit extinction spectra with a single peak at a wavelength of ∼540 nm. This indicates the presence of a single LSPR mode in each sample at a similar excitation wavelength. The sharpness of the peaks further confirms the uniform sizes of the Au nanoparticles, consistent with the observations from SEM and HRTEM.

### Electrocatalytic performance of Au NPs

$CO_2RR$ is considered a promising solution for reducing excessive $CO_2$ in the atmosphere. However, the major hurdles in $CO_2RR$ lie in its high activation energy and limited selectivity. Recognizing the potential of Au as a catalyst with low activation energy and high selectivity, we conducted measurements to evaluate the electrocatalytic performance of Au NPs with three different exposed facets (for the details of our experiments, see Methods section, Scheme S1, Figs. S3, S4). As shown above, we have synthesized three uniform Au nanostructures NCs {100}, RDs {110}, and OCs {111} with the same phase, similar sizes, and LSPR wavelengths. The corresponding current−time ($i$−$t$) curves, linear sweep voltammetry (LSV) curves and the corresponding Tafel plots are presented in Fig. S5 and Fig. S6, respectively. Under all applied potentials (from −0.53 $V_{RHE}$ to −0.88 $V_{RHE}$ with the step of −0.07 $V_{RHE}$), the major gas products detected were carbon monoxide and hydrogen. Liquid products were traced by proton nuclear magnetic resonance ($^1$H NMR) and no products were detected (see Fig. S7). Control experiments with carbon powder deposited on carbon paper also confirmed that $CO_2RR$ is catalyzed by the Au NPs instead of the carbon support (Fig. S8). Experiments conducted with Ar bubbling further validate that the generated CO originates solely from the reduction of $CO_2$ rather than from ligand decomposition, as depicted in Fig. S14.

The FE(CO), FE($H_2$) and the corresponding partial current density of Au NCs, RDs, and OCs are shown in Fig. 2a and S9. The electrocatalytic $CO_2RR$ selectivity of the three Au NPs differs significantly in between them. Au RDs with {110} facet demonstrated the highest FE(CO) at all applied potentials from −0.53 to −0.88 $V_{RHE}$, achieving its maximum of FE(CO) ≈ 94% at −0.67 $V_{RHE}$. In comparison, NCs {100} presented a lower CO selectivity at the same applied potential (FE(CO) ≈ 69%) and OCs {111} showed the lowest among three Au NPs with FE(CO) ≈ 51% at −0.67 $V_{RHE}$. The long-term performance ($i$−$t$ curves) for $CO_2RR$ on the three Au NPs was also checked (Fig. S13), resulting in the same order for the FE(CO) performance after 5 h: RD > NC > OC, indicating the high stability of the system.

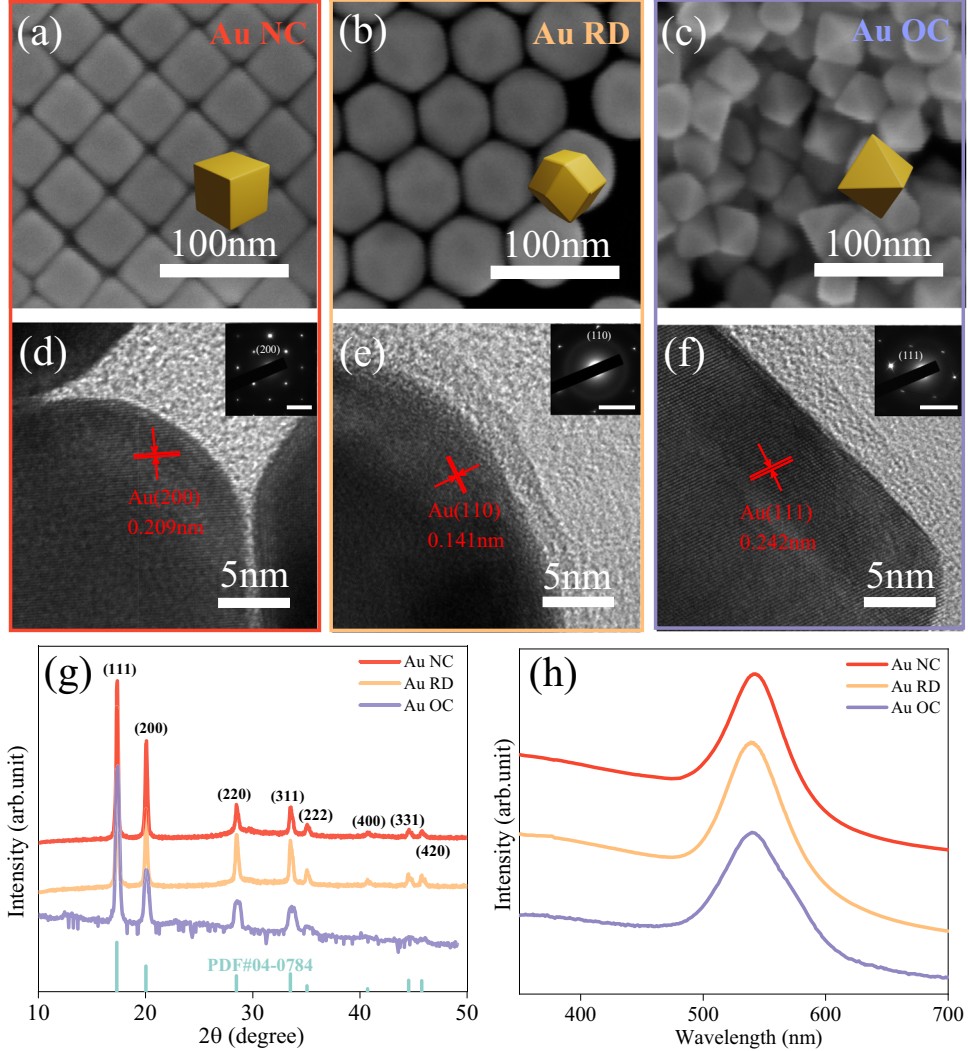

**Fig. 1 | Characterization of Au NPs. a–c** SEM images of (**a**) Au NCs, (**b**) RDs, and (**c**) OCs. **d–f** HRTEM and SAED (inset) patterns of (**d**) Au NCs (**e**) RDs, and (**f**) OCs, the scale bars in insets are 5 1/nm. The average distances between fringes and the corresponding Au facets are marked in red. **g** XRD patterns of Au NCs (red), RDs (yellow), OCs (purple) and expected XRD pattern for Au in the FCC phase for Mo Kα radiation. **h** Extinction spectrum of Au NCs, RDs, and OCs with LSPR peaks at 543 nm, 539 nm, and 542 nm, respectively. Source data are provided as a Source Data file.

DFT calculations were performed to elucidate the reasons behind the different electrocatalytic $CO_2$ reduction performance among the three Au nanoparticles. The calculation details are described in Methods. The calculations revealed that the activation of $CO_2$ is the rate-determining step, as depicted in Fig. S10. The formation energy of COOH* on Au{110} (RDs) is 1.1 eV, which is lower than 1.2 eV on Au{100} (NCs) and 1.4 eV on Au{111} (OCs). The lowest COOH* formation energy on Au{110} (RDs) facilitates the $CO_2$ molecule activation and the CO generation. In summary, regarding FE(CO), the electrocatalytic performance of Au NPs with different exposed facets can be ordered as Au RD {110} > NC {100} > OC {111}, which is a consequence of the different energy barriers observed on the corresponding facets. The results are also consistent with previous works based on different Au facets in dark conditions[18].

### Plasmon-assisted electrocatalytic performance of Au NPs with different exposed crystal facets

As mentioned above, plasmon excitation has recently emerged as a new way of exciting photoactive materials that may help improve the electrocatalytic selectivity and current density of a wide range of reactions[56]. To investigate how plasmons affect our system, the plasmon-assisted electrocatalytic $CO_2RR$ on Au NPs was subsequently studied. A Light-emitting diode (LED) with a wavelength of 525 nm was chosen as the illumination source because this wavelength is close to the LSPR maxima of the Au NPs. The cathode was continuously illuminated by the LED (610 mW/cm²) in this system. The corresponding $i–t$ and LSV curves are presented in Fig, S5 and Fig, S6, respectively. Similar to the electrocatalytic $CO_2RR$, the major gaseous products in plasmon-assisted electrocatalysis were CO and $H_2$, and no liquid products were detected by ¹H NMR, as shown in Fig, S7. Control experiments with carbon powder were also done to eliminate the influence of carbon, as shown in Fig. S8.

Plasmons showed a significant impact on the catalytic selectivity and activity of Au NPs, as shown in Fig. 2b, c. While Au OCs display the lowest FE(CO) in (dark) electrocatalytic $CO_2RR$, the excitation of plasmons notably enhanced their selectivity. Illumination resulted in an increase of at least 19% in the absolute value of FE(CO) on Au OCs compared to pure electrocatalysis at all tested potentials. Notably, the FE(CO) for Au OCs nearly doubles (43%) upon plasmon excitation at −0.81 $V_{RHE}$; where it increased from 44% in the dark to 87% under illumination. When illuminated, Au NCs also showed an FE(CO) enhancement, ranging between 12% and 27% at all applied potentials. In contrast, Au RDs, which had the highest FE(CO) in pure

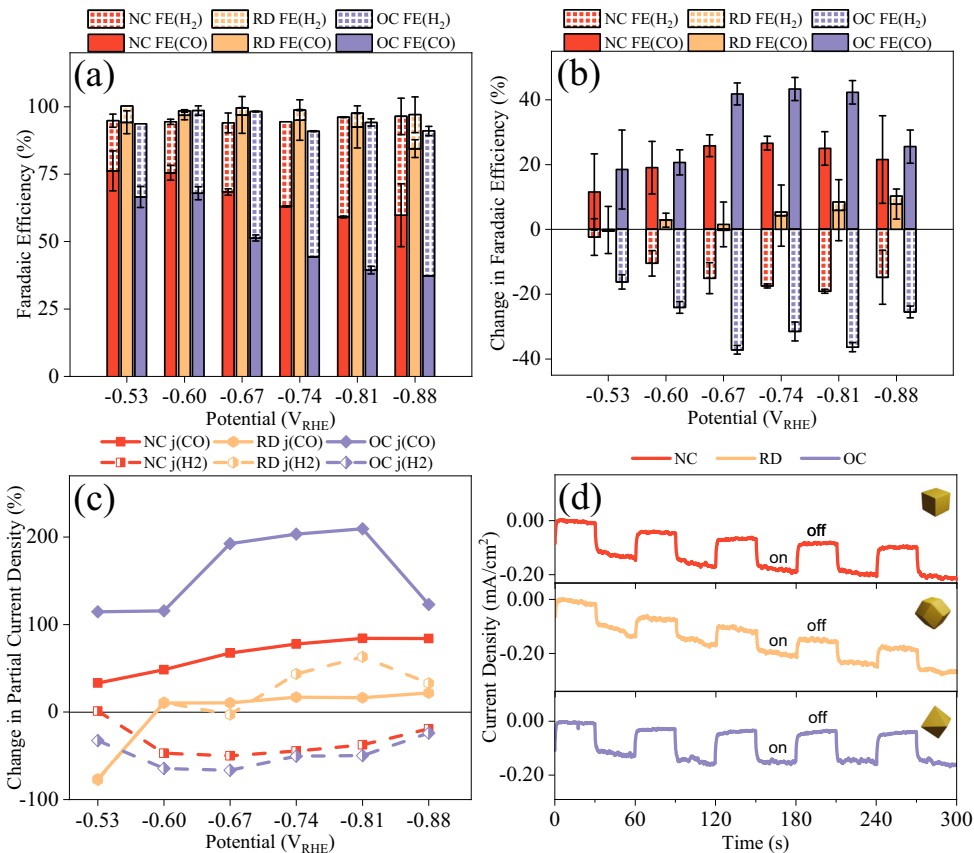

**Fig. 2 | (Plasmon-assisted) electrocatalytic CO₂RR performance of three Au NPs.** **a** Faradaic efficiency (FE) of CO (solid) and H₂ (square grid) production on Au NCs (red), RDs (yellow) and OCs (purple) in an electrocatalytic CO₂ reduction system. The error bars denote the standard deviation obtained from three independent measurements. **b** Change in the absolute value of FE for CO and H₂ [FE_light - FE_dark] on Au NPs when illuminated by a 525 nm LED. **c** Percentage change in partial current density for CO (solid line) and H₂ (dotted line) [(j_light·j_dark)/j_dark × 100%] on Au NPs when illuminated. **d** Chopped-light chronoamperometry with 525 nm LED at −0.81 V_RHE on Au NPs. Source data are provided as a Source Data file.

electrocatalytic CO₂RR, showed minimal improvement when illuminated. More specifically, under most applied potentials, the increase of FE(CO) was less than 5%. In parallel, the FE(H₂) notably diminishes on both Au OCs and NCs, as depicted in Fig. 2b. This decline serves as a clear indicator of the suppression of the competitive HER reaction. Conversely, Au RDs show a marginal increase in FE(H₂) at elevated potentials; however, it is important to note that this enhancement is significantly less pronounced compared to the concurrent increase observed in FE(CO).

The partial current density of CO (j(CO)) on Au OCs and NCs also showed noticeable increase with illumination as shown in Fig. 2c. Specifically, at −0.81 V_RHE, the j(CO) on Au OCs experienced a remarkable enhancement of 210% upon plasmon excitation compared to the pure electrocatalytic process. Similarly, Au NCs displayed an improvement of 84% at the same potential upon introducing plasmons. However, Au RDs only exhibited a modest increase of 16% in the presence of light. The chopped *i–t* curves for three Au NPs are presented in Fig. 2d, confirming the noticeable increase in the current induced by the light. Au OCs and NCs showed pronounced plasmon-induced photocurrents of 0.12 and 0.11 mA/cm², respectively, indicating a large photo-response. In contrast, Au RDs presented a lower photocurrent response of 0.08 mA/cm². In summary, the introduction of plasmons during electrocatalytic CO₂RR led to substantial enhancements in activity and selectivity to CO, both on Au OCs and NCs. However, Au RDs, despite possessing the highest intrinsic (i.e. dark) electrocatalytic performance, exhibited a poor response to light and showed only minimal improvement.

## Study of the mechanism behind plasmon-enhanced electro-catalytic CO₂RR

We now turn to understand the underlying mechanism behind plasmon excitation on the different Au NPs electrocatalytic systems. After LSPR is triggered, the non-radiative decay of the plasmon results in the generation of energetic "hot" electrons and holes[61–63]. These hot carriers can then be transferred to adsorbed molecules, facilitating catalytic processes on the surface of the nanoparticles. Additionally, the energy carried by the hot carriers can also be transferred to other electrons through electron–electron collisions and to the lattice via electron–phonon interactions, leading to an increase in the surface temperature[64]. Given the exponential increase of reaction rates when increasing the temperature (Arrhenius law), we first focused on investigating the impact of heat as the main channel to explain the enhanced activity and selectivity reported in the previous section.

To explore the role of heat in our systems, we conducted CO₂RR electrocatalytic experiments in the dark under three different temperature conditions: room temperature (23 °C), 30 °C, and 40 °C. We achieved these temperatures by means of external heating with a hot plate. In contrast to the scenario when plasmons are excited, the temperature-dependent experiments revealed an inverse relationship regarding FE(CO), as depicted in Fig. 3a–c. As the temperature increased, there was a noticeable decline in selectivity towards CO, favoring the HER at higher temperatures. For instance, at −0.67 V_RHE, the FE(CO) on Au RDs decreased from 94% at room temperature to 51% at 30 °C, and further dropped to 13% at 40 °C. This trend was consistent across all three Au NPs, indicating a clear reduction in selectivity towards CO at higher

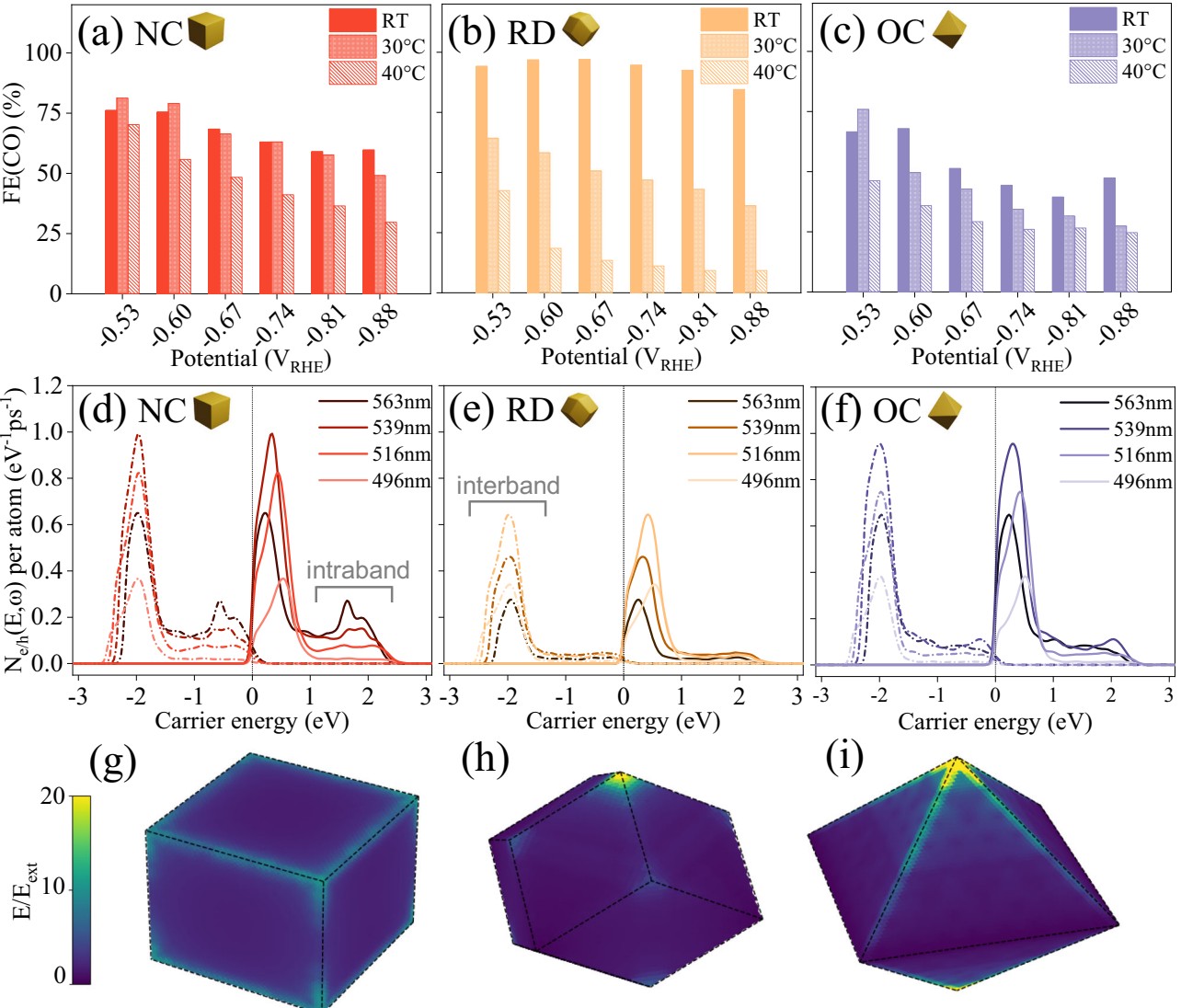

**Fig. 3 | Mechanism investigation of plasmonic electrocatalytic CO₂RR based on three Au NPs. a–c** Faradaic efficiencies (FE) for CO production at different applied potentials on (**a**) Au NCs, (**b**) RDs, and (**c**) OCs at room temperature (solid), 30 °C (dense slash) and 40 °C (sparse slash). **d–f** Hot electron (solid) and hole (dashed) generation rate for each of the (**d**) Au NCs, (**e**) RDs, and (**f**) OCs at different electric field wavelengths. $N_{e/h}(E,\omega)$ indicates the number of hot carriers being generated with energy $E$ upon the light with $\omega$. The Fermi energy is set to zero. **g–i** Absolute electric field profile in (**g**) Au NCs, (**h**) RDs, and (**i**) OCs at the corresponding LSPR frequencies, in reference to the external applied electric field. All simulated nanoparticles have approximately 200,000 atoms. Source data are provided as a Source Data file.

temperatures. We then estimated the temperature on the surface of the electrode under plasmon excitation to be ~70 °C[65]. At this high temperature, the performance of the CO₂ reduction in the dark is even worse, as shown in Fig. S17. This could be in part associated with the reduced solubility of CO₂, which drops nearly 4 times when the temperature increases from 20°C to 70°C. As such, the enhancement of FE(CO) under plasmon excitation has an even larger impact than expected, as it needs to counterbalance the negative impact of fewer CO₂ molecules dissolved in solution. To have real dimension, a proper comparison would be for instance, at −0.81V_RHE the FE(CO) on Au OCs under plasmon excitation is 87% (Fig. 2), while in dark but under a similar surface temperature of around 70°C the FE(CO) is only 5% (Fig. S17). Hence, while local heating plays a significant role in various LSPR-involved systems, it has the exact opposite trend here reinforcing the non-thermal effects associated in our study. This finding highlights the unique properties and advantages of plasmonic catalysis compared to

conventional thermal catalytic processes, as the enhancement of selectivity to CO cannot be achieved thermally[66,67].

As heat is not responsible for the observed effects under illumination, our study then focused on the impact of hot carriers generated after plasmon dephasing. Due to the high energy of hot carriers and their ability to be transferred to adsorbed molecules, they can play critical roles in plasmonic catalytic processes. Relevant to our study is the well-known sensibility of hot carriers on the geometry of the plasmonic catalyst[68,69]. Moreover, hot carriers can induce processes like assisted desorption of CO from the surface, enabling further CO₂ molecules to adsorb on CO-active sites, leading to an increase in current density and potentially also in the FE(CO). They could also participate in other processes, such as the activation of adsorbed CO₂. In order to explore the potential role of hot carriers in our system, we calculated the hot-carrier generation rate in Au NCs, OCs, and RDs. To accomplish this, we employed the recently developed approach of Jin and coworkers[70]. In this method, the electric potential induced by the

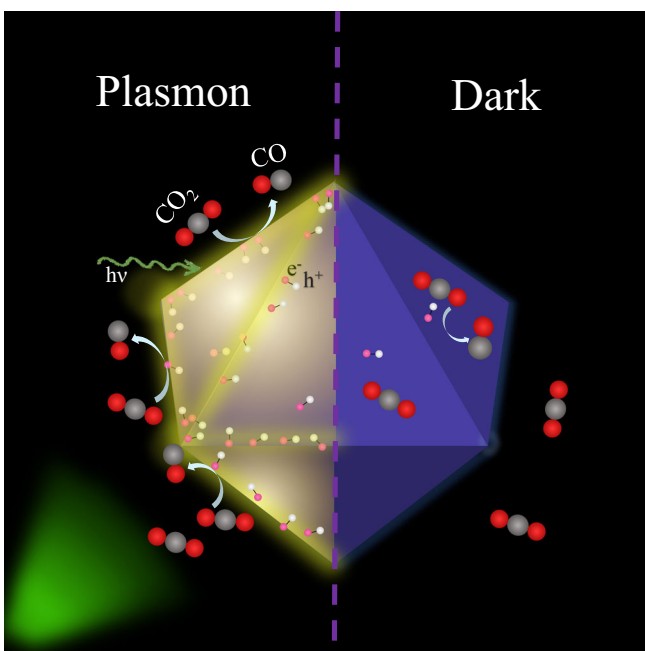

**Fig. 4 | Schematic diagram of the electrocatalytic CO₂RR response of a AuNP under illumination (i.e. plasmon excitation) versus its reactivity under dark conditions.** While the response is dominated by the exposed crystal facets in dark, the FE(CO) highly enhances upon plasmon excitation and the significance of uncoordinated sites (i.e. edges, corners) appears to eclipse that of facets in the overall catalytic response.

light is first calculated using the quasistatic approximation and then used to evaluate Fermi's golden rule (FGR). The FGR is evaluated using large-scale atomistic tight-binding simulations in which each nanoparticle consists of approximately 200,000 Au atoms (see Methods for more details). Figure 3d–f shows the distribution of hot electrons and hot holes, exhibiting pronounced peaks associated with interband transitions from states with d-band character to states with sp-band character. In addition, smaller peaks (i.e. clearly visible for the Au NCs) are caused by intraband transitions involving both initial and final states with sp-band character. These peaks arise from surface-enabled transitions, which get magnified due to the field enhancement at the surface of the nanoparticle.

Our results demonstrate that Au RDs generate fewer hot carriers compared to Au NCs and OCs, while the latter two exhibit similar levels of hot carrier generation. Notably, the hot carrier generation is strongly dependent on the electric field enhancement on the surface of plasmonic nanomaterials[68]. Moreover, the direct transfer of hot electrons generated by LSPR is strongly influenced by the local electric field conditions[71]. Consequently, we conducted an analysis of the electric field distribution near the surface of Au NPs, as depicted in Fig. 3g–i. Notably, Au OCs and NCs exhibit more prominent electric field enhancements on their edges and corners compared to Au RDs. For this reason as well, the Au NCs generate the largest portion of intraband transitions (from 10% of total transitions at 496 nm to 32% at 563 nm) compared to the Au RDs (6% to 15%) and the Au OCs (7% to 17%). For the purpose of this counting, electrons are counted as coming from intraband transitions if their energies are above 1.2 eV, the specific values of population are demonstrated in Table S5. Large-scale atomistic simulations and electromagnetic modeling, in conjunction with experimental evidence of weaker light response in RDs compared to NCs and OCs, reinforce the significance of hot carriers and electric field enhancement in plasmonic catalytic processes.

Interestingly, facets can mostly account for the dark electrocatalytic performance of Au OCs, NCs, and RDs in the CO₂RR. However, uncoordinated sites (edges) are the main responsible sites for the plasmonic-assisted CO₂RR in these systems. This is a remarkable result,

as it highlights that plasmonic catalytic sites (or reactive hot spots) may differ from the inherent surface expected sites in the dark[46,72].

The plasmon-induced electric field enhancement is known to be highly dependent on the exact geometry of the nanostructure[42]. Remarkably, on our three different Au NP geometries, pronounced enhancements were observed at low-coordinated sites, specifically corners and edges, rather than facets. This is consistent with previous investigations into the atomic structure of plasmonic nanoparticles, which revealed a non-uniform spatial distribution of hot electrons: they tend to concentrate more prominently at lower-coordinated sites such as edges, as opposed to higher-coordinated sites such as facets[73]. A recent study applying single-particle electron energy loss spectroscopy also revealed the spatially inhomogeneous carrier extraction efficiency in plasmonic nanostructures[69]. Notably, investigations into the impact of low-coordinated sites have established their superiority over facets in catalytic processes[74], including the CO₂RR system[75,76]. However, as shown in Fig. 2, in our system facets dominate the dark response, while under light activation the energetics of the reaction is dominated by edges and corners. In order to understand this change in reactivity, we conducted DFT simulations to study the CO₂RR reactivity of edge and tip sites, as shown in Fig. S10. Interestingly, the activation energy on the edges (0.65 eV) was found to be similar to the value on the Au{110} facet exposed by Au RDs (0.64 eV). The similarity in energy barriers between the Au{110} facet and the edges implies that the plasmon-induced hot electrons generated on the edges, have a limited impact on improving FE(CO). Yet the values on Au OCs {111} (0.74 eV) and NCs {100} (0.68 eV) are higher than those on the edges (0.65 eV). Consequently, for Au NCs and OCs, edges with low coordination number—once activated with light—serve as the pivotal juncture where highly active CO₂RR sites coincide with high electric fields and a substantial population of hot electrons. Thus, plasmon excitation can make a difference in the reactivity of these systems by enabling active sites with lower activation barriers than in dark. The efficient transfer of hot electrons to the adsorbed CO₂ molecule facilitates its activation, resulting in a significant enhancement in the activity and selectivity of CO₂RR. Even though these sites are also present in the dark experiments, our results indicate that they become much more active—even

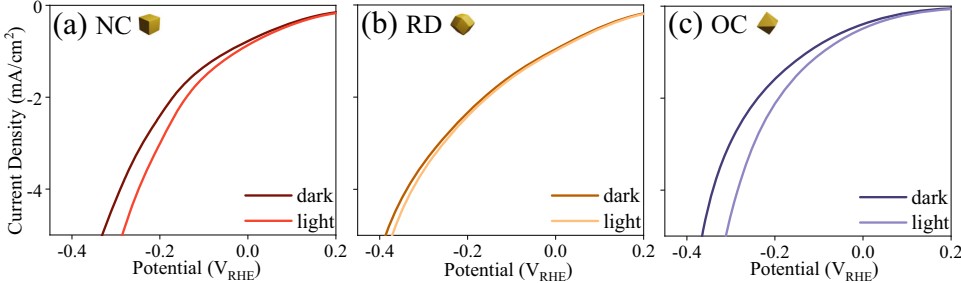

**Fig. 5 | HER performance on Au NPs.** LSV curves of (**a**) Au NCs, (**b**) Au RDs, and (**c**) Au OCs in dark and with 525 nm LED illumination. Source data are provided as a Source Data file.

dominating the overall response of the system—when the plasmon is excited. The mechanism is depicted in Fig. 4. As such, Au NPs with a higher edge/facet ratio, such as Au OCs (as supported by calculations and specific values provided in Table S4), are expected to exhibit the most pronounced plasmonic response regarding $CO_2RR$, as corroborated by experimental observations.

To comprehensively investigate the mechanism, the influence of light wavelength on plasmonic catalytic performance was then examined. An LED with a wavelength of 405 nm, operating at the same power as the 525 nm LED (610 mW/cm²), was employed for this purpose. Despite the higher energy of 405 nm photons compared to 525 nm photons, the enhancement observed in FE(CO) was smaller, particularly evident in the case of Au NCs (see Fig. S11). To investigate the underlying reasons for this observation, we conducted calculations of energy absorption per atom for each nanoparticle shape and region as a function of frequency, as depicted in Fig. S16. The key disparities arise from the edges, where absorption changes significantly with varying wavelengths. Specifically, NCs display the most substantial change, with absorption decreasing by 70% as the wavelength shifts from 540 nm to 413 nm—a decrease exceeding 2000 eV/ns. In comparison, OCs decreased by 1140 eV/ns, and RDs decreased by 180 eV/ns. This large variation in NCs may be attributed to the reduced intraband transition, as illustrated in Fig. 3d–f, where the population decreased from 32% at 563 nm to 10% at 496 nm. Furthermore, simulating electric potential enhancement under 405 nm illumination revealed a more pronounced effect on the facets for NCs (Fig. S16). Consequently, fewer hot electrons are generated and tend to distribute on sites with higher energy barriers. Thus, compared to 525 nm illumination, fewer $CO_2$ molecules are activated, resulting in a significantly smaller improvement in FE(CO). These findings contribute additional evidence underscoring the significance of hot electron generation and high electric field enhancement in our model system.

To verify the validity of our hypothesis regarding the prominence of low-coordinated sites with high electric field and a substantial quantity of hot electrons over facets in diverse plasmonic catalytic systems, we conducted experiments utilizing the same Au NPs for the hydrogen evolution reaction (HER), details of experiments are shown in Methods section. Under dark conditions, the HER onset potentials on Au NPs are in the order of: RD{110} <NC{100} <OC{111}, as shown in Fig. 5a–c, which is consistent with the DFT simulation results (Fig. S12). With illumination, Au OCs and NCs, which possess a large number of hot carriers and exhibit higher electric field enhancements, as shown in Fig. 3, show a remarkable reduction in HER onset potential by 0.033 $V_{RHE}$ and 0.021 $V_{RHE}$, respectively. In contrast, Au RDs only present a slight improvement of 0.007 $V_{RHE}$ (Fig. 5a–c). The corresponding Tafel plots are presented in Fig. S15. To assess the statistical robustness of our results, we performed each experiment three times. Across each round of experiments, the Au NPs consistently demonstrated the same trend in the change of onset potentials, as depicted in Fig. S15. The remarkable plasmon interaction observed in Au OCs and NCs, coupled with the limited response in HER exhibited by Au RDs, provides additional support for the significance of abundant hot carriers and intense electric field enhancement in plasmonic catalytic systems. The largest reduction in onset potential observed in OCs, which possess a high edge/facet ratio, again emphasizes the dominance of low-coordinated sites over facets in at least the two investigated systems here.

In summary, contrary to electrocatalytic systems, the role of facets seems not to be prominent in plasmonic electrocatalysis. Instead, low-coordinated sites, such as edges with significant electric field enhancement along with a high distribution of hot electrons, play more significant roles. These results, which deviate from conventional catalytic processes, exhibit a unique signature of plasmonic catalysis.

## Discussion

We studied the effect of plasmon excitation in electrocatalytic $CO_2RR$ and HER systems based on three Au NPs morphologies: Au NCs, RDs, and OCs. In pure electrocatalytic $CO_2RR$, Au RDs showed the best FE(CO), with values as high as 94% at −0.67 $V_{RHE}$, NCs and OCs presented lower selectivity, with 69% and 51%, respectively. These differences primarily stem from the distinct formation energy of COOH* on the specific exposed Au facets, confirming the importance of facets in pure (i.e. dark) electrocatalysis[18]. However, when introducing plasmons into the system, Au OCs—which had the lowest CO selectivity in electrocatalytic $CO_2RR$—presented the most significant enhancement: at −0.81 $V_{RHE}$ the FE(CO) with illumination was twice as high as the FE(CO) in dark conditions, resulting in an increase from 44% to 87%. Au NCs also showed an FE(CO) increase of 26% under the same conditions. In contrast, Au RDs demonstrated a modest increase of only 5% in FE(CO). Moreover, at this potential Au OCs exhibited a remarkable 210% enhancement of j(CO) and NCs also showed an 84% improvement, yet RDs only demonstrated a slight improvement of 16% upon plasmon excitation. These findings underscore the substantial influence of plasmons on selectivity and activity, with distinct variations observed among different catalysts.

To further investigate the underlying mechanism behind the divergent responses to plasmonic effects, we initially examined the impact of temperature. Our control experiments clearly show that FE(CO) decreases when the temperature increases for our Au catalysts, which implies that plasmonic heating does not play a role in the improved FE(CO) performance of the Au catalysts under illumination. This highlights the existence of a different mechanism for plasmonic catalysis compared to thermal catalysis. Trying to rationalize these results, our focus shifted toward exploring the generation of hot carriers. Based on large-scale atomistic simulations of hot carrier generation and electromagnetic field modeling of the Au NPs, we found that Au OCs and NCs generated more hot carriers and stronger electric field enhancement than Au RDs. These findings align with our experimental results of stronger light response of Au OCs and NCs than RDs, underscoring the significance of hot carriers and electric field in this plasmonic electrocatalytic processes. Furthermore, electromagnetic field modeling revealed significantly stronger enhancements at low

coordinated sites such as corners and edges, suggesting a higher concentration of hot carriers at these locations compared to facets. Moreover, previous studies have also demonstrated the catalytic favorability of edges in $CO_2RR$[75,76].

Based on these observations, we proposed a mechanism for the plasmon-enhanced $CO_2RR$ performance: low coordinated sites on nanostructures concentrate the electric field more intensely, resulting in a higher abundance of hot carriers. These abundant hot carriers on the edges facilitate the activation of $CO_2$ molecules, leading to enhanced selectivity and activity. In contrast, facets do not play a significant role in this process. This is in line, for example, with a recent pollution degradation study based on Ag octahedron nanoparticles modified with $Cu_2O$ - in which the edge-covered NPs demonstrated higher catalytic activity than the fully covered ones - supporting our theory on the dominant role of edges in plasmonic catalysis[77]. To validate our hypothesis regarding the greater importance of low-coordinated sites compared to facets in diverse plasmonic catalytic processes, a study based on a different reaction - HER - was then carried out. Au OCs and NCs, which generated a larger number of hot carriers and located at edge sites with significantly enhanced electric fields, exhibited notable enhancements in catalytic activity upon plasmon excitation. On the other hand, RDs with fewer hot carriers displayed a weaker plasmonic response, providing further confirmation of our proposed mechanism. As such, this study not only delves into the distinctive reactivity of plasmonic catalysts but also furnishes guidance for crafting the next iteration of light-activated catalysts.

## Methods

### Synthesis of Au NPs

Chemicals used were ascorbic acid (>99.0%), hexadecyltrimethylammonium bromide (CTAB, > 96.0%), hexadecyltrimethylammonium bromide (CTAB, > 99.0%), gold (III) chloride trihydrate ($HAuCl_4 \cdot H_2O$, > 99.9%), Sodium borohydride (>98.0%), sodium hydroxide (>97%), sodium carbonate (99.999%), perchloric acid (70%). All chemicals were used after purchase without any further purification. Milli-Q water at 25 °C was used in all experiments. All glassware was washed with *aqua regia*, Milli-Q water, and dried before use. Carbon (mesoporous, nano powder graphitized 99.95%, trace metals basis), nafion (perfluorinated resin solution, 5 wt% in lower aliphatic alcohols and water, contains 15-20% water), 2-Propanol (puriss.p.a., ACS reagent. reag. ISO, reag. Ph. Eur., > 99.8% (GC)), deuterium oxide (deuteration degree min. 99.9% for NMR spectroscopy) and dimethyl sulfoxide (ACS reagent >/=99.9%) were also purchased from Sigma-Aldrich. Carbon paper (sigracet 39BB and sigracet 39CC) was purchased from Fuel cells, etc.

Au NCs. Au NCs, and Au RDs were synthesized in accordance with the literature[59]. Firstly, 0.6 mL 10 mM ice-cold $NaBH_4$ was rapidly injected into 10 mL pre-prepared aqueous solution containing 0.25 mM $HAuCl_4$ and 75 mM CTAB. Then the brown solution was kept stirring slowly at 30 °C for 2 h and diluted 100 times with DI water and used as seed hydrosol. 0.3 mL seed hydrosol was then added into 25 mL growth solution containing 0.04 mM $HAuCl_4$, 16 mM CTAB and 6 mM ascorbic acid and mixed by a vortex-mixer for 10 s. The reaction mixture was left undisturbed at 25 °C overnight and seed preparation was finished. For the growth of Au NCs, 0.150 mL of 25 mM $HAuCl_4$ was quickly added into 8 mL seed solution, and left undisturbed at 30 °C for 2 h after being mixed by a vortex-mixer for 10 seconds. The red-purple color indicated the formation of Au NCs.

Au RDs. 0.5 mL of 0.1 M ascorbic acid, 0.55 mL of 0.2 mM NaOH and 0.2 mL 25 mM $HAuCl_4$ were added into 8 mL above synthesized Au NCs solution in order. The whole solution was left undisturbed at 30 °C for 2 h untill a red-brown Au RDs colloidal was obtained.

Au OCs. Au OCs were synthesized in accordance with the literature[60]. Au octahedral seeds were firstly prepared: 0.6 mL

10 mM ice-cold $NaBH_4$ solution was injected into the mixture of 87.5 μL of 20 mM $HAuCl_4$ solution and 7 mL of 75 mM CTAB solution. After 3 h of gentle stirring, the mixture was diluted 100-fold with DI water to get the octahedral seed hydrosol. 0.15 mL seed hydrosol was added to the growth solution containing 25 μL of 20 mM $HAuCl_4$, 0.387 mL of 38.8 mM ascorbic acid and 12.1 mL of 16 mM CTAB solution, and mixed by the vortex mixer thoroughly, then left unperturbed at 25 °C overnight. Next, 5 mL previously obtained solution was added into 12.5 mL of second growth solution containing 16 mM CTAB, 0.04 mM $HAuCl_4$ and 1.2 mM ascorbic acid and kept undisturbed overnight. The purple color of the solution indicated the formation of Au OCs.

### Characterization of Au NPs

TEM and HRTEM measurements were carried out on JEM1011 operated at 80 kV and JEM-2100F operated at 200 kV, respectively. SEM was carried out on Ultra plus scanning electron microscope (Zeiss) with 10 kV acceleration voltage. UV–vis–NIR absorption spectra were measured on UV–vis–NIR spectrometer Lambda 750 (Perkin Elmer). XRD was carried out with a custom-built molybdenum Kα X-ray reflectometer/diffractometer. The samples were measured at a fixed incidence angle of 10°, and intensities at different scattering angles (2θ) were recorded from 10 to 60° in 2500 steps (0.02°/step) for 10 s each using a point detector (NaI scintillation counter). EA (C, H, N) test was measured with the Heraeus Elementar Vario EL instrument.

### $CO_2RR$ (plasmonic) electrocatalytic experiments

**Preparation of working electrodes.** Au NPs and carbon powder were mixed with an Au mass loading of 20 wt% to avoid aggregation of Au NPs. The mass ratio was also confirmed by the elemental analysis (EA) of C, H, N, as shown in Table S2. Then 5 mg Au/C composite then was dispersed in a mixture of 0.5 mL 2-propanol, 0.5 mL water and 10 μL nafion to obtain the 'ink'. Au/C composites were measured by TEM and confirmed that Au NPs were evenly dispersed in porous carbon, as shown in Fig. S3. Carbon paper was cut in the size of 0.5 cm × 2 cm. Then 40 μL ink was dropped on carbon paper in an area of 0.5 cm × 1 cm to obtain the working electrode. Therefore, a mass loading of Au/C component on each electrodes should be 200 μg with 40 μg Au NPs inside. For the control experiment with pure carbon, the mass loading of carbon powder should be 200 μg.

**$CO_2RR$ (plasmon-assisted) electrocatalysis measurement.** $CO_2RR$ measurement was carried out in a customized H-type cell connected with a potentiostat and an online gas chromatograph (Clarus 590 GC, Perkin Elmer, FID, and TCD detectors), as shown in Scheme S1. A carbon paper loaded with Au/C composite is applied as the working electrode, as mentioned above. A platinum mesh and an Ag/AgCl electrode were introduced as a counter electrode and reference electrode, respectively. 0.1 M $NaHCO_3$ was chosen as the electrolyte and was constantly purged with $CO_2$ (99.998%, Linde) with a rate of 30 sccm controlled by a flow meter (PR4000B, MKS). The gas flow rate after passing through the GC was measured as relatively smaller than the gas inlet, therefore the values of Faradaic Efficiency in this study were normalized by dividing a value of 0.9. Before the experiment, the electrolyte was saturated with $CO_2$ by bubbling $CO_2$ gas for at least 1 h. During the measurement, the gas products from the cell were injected online every 30 min into the GC, while the liquid products were detected by proton nuclear magnetic resonance ($^1H$ NMR) of the electrolyte after the reaction was completed. Before the measurement, the electrochemically active surface areas (ECSA) of working electrodes were measured. The method to determine the ECSA and the measured values are presented in S4, Fig. S4 and Table S3. The current density shown in this work is obtained by dividing the total current by the corresponding ECSA. Except for temperature-dependent measurements, all experiments were conducted at room temperature (23 °C).

For photo-electrocatalytic measurement, high-power LEDs with 525 nm and 405 nm wavelength (Mightex Systems) were used as the illumination source. The LEDs were first focused by a lens, then mounted in front of the reactor with a distance of about 5 cm. The power on the front side of the reactor was measured as 610 mW/cm² by the power meter.

### Detailed measurement parameter setting

**Gas Chromatography.** Gas chromatography (GC) (Arnel Engineered Solutions Clarus 590 GC from Perkin Elmer company) with hydrogen generator (NM plus from Perkin Elmer) was connected with reactor cells for online measurement of gaseous products. The carrier gas of GC was Argon (Ar) with a flow rate of 27.0 mL/min. The injector temperature was set to 400 °C. The GC was equipped with the thermal conduction detector (TCD) and flame ionization detector (FID). The parameters of the two detectors were set as follows: for TCD, the temperature was 200 °C. For FID, $H_2$ and the air were blown into the detector with the flow rate of 45.0 mL/min and 450.0 mL/min, respectively, the temperature was 250°C. For the column, the temperature was set to 70 °C for 5 min at first, then heated up to 150°C at a speed of 20 °C/min and maintained for 17 min.

**[1]H NMR.** To detect the liquid products of $CO_2RR$, 0.9 mL electrolyte after reaction, 0.1 mL Deuterium oxide and 1 μL Dimethyl sulfoxide were thoroughly mixed and injected into the NMR tube for [1]H NMR measurement. The [1]H NMR spectra were recorded on Avance III HD 500 MHz Bruker BioSpin spectrometers.

**Electrochemical tests.** Several electrochemical measurements were carried out, including cyclic voltammetry (CV), open-circuit potential (OCPT), Electrochemical impedance spectroscopy (EIS), linear sweep voltammetry (LSV) and current–time curve ($i$–$t$). The CV was sent to sweep from −0.6 V to −2.0 V with 20 segments, and the sample interval was 10 mV. For LSV, the initial and final potential as well as the interval were the same as for CV. EIS was conducted to measure the solution resistance to do the iR drop compensation, and its testing potential was set as the open circuit potential obtained from the OCPT measurement. The frequency was scanned from $10^5$ Hz to 1 Hz. For all $i$–$t$ curves, the measurements were carried out for 3600 s. All the potentials were calibrated and converted to a reversible hydrogen electrode (RHE).

### Electrochemically active surface area measurement

The specific electrochemically active surface area (ECSA) of electrodes was treated as the area of electrodes[78]. In our study, ECSA was determined by integrating the cathodic peak in cyclic voltammetry (CV) curves, as shown in Fig. S4. ECSA of electrodes was estimated with CV sweeping between 0 and +1.5 V with a scan rate of 10 mV/s in the electrolyte of 0.1 M $HClO_4$ solution. The peak areas of CV curves were integrated and then divided by the scan rate with 450 μC/cm² (the specific charge required for gold oxide reduction)[79]. The specific ECSA of electrodes is shown in Table S3.

### DFT Calculation

To explore the mechanisms of $CO_2$ reduction to CO on different Au facets, 4 × 2 Au (100), Au (110), and Au (111) periodic surface slabs including four atomic layers were built as shown in Fig. S10a–c. A vacuum slab with 30 Å was added to avoid the interaction influence of the periodic boundary conditions. Each model contains 128 atoms. To explore the $CO_2RR$ reactivity of edge and tip sites, two models of Au nanoclusters with Au {100} and Au {111} as mainly exposed facets were developed, respectively, as shown in Fig. S10d, e. The Au nanocluster models contain 147 and 85 Au atoms, respectively.

The DFT calculations were performed by VASP with the projector augmented wave (PAW) method[80,81]. The exchange and correlation

potentials are present in the generalized gradient approximation with the Perdew–Burke–Ernzerhof (GGA-PBE)[82,83]. The 1 × 2 × 1 k-points were used for the Brillouin zone integration. The cutoff energy, the convergence criteria for energy and force were set as 450 eV, 1 × 10⁻⁵ eV/atom and 0.02 eV/Å, respectively.

The mechanism of $CO_2$ reduction to CO is considered as

$$CO_2 + ^* + H^+ + e^- \rightarrow {}^*COOH \tag{1}$$

$$^*COOH + H^+ + e^- \rightarrow {}^*CO + H_2O \tag{2}$$

$$^*CO \rightarrow CO + {}^* \tag{3}$$

The asterisk (*) of above means the substrate.

The change of Gibbs free energy (ΔG) for each reaction step is given as follows[84]:

$$\Delta G = \Delta E + \Delta ZPE - T\Delta S \tag{4}$$

where ΔE represents the total energy difference between the product and the reactant. ΔZPE and TΔS are the zero-point energy correction and the entropy change at 298.15 K, respectively.

### HER (plasmonic) electrocatalytic experiments

HER measurement was carried out in a typical three-electrode cell, as shown in Scheme S2. The electrodes were the same as for $CO_2RR$. 0.1 M $HClO_4$ solution was chosen as the electrolyte and was purged with $N_2$ for 30 min to remove the $O_2$ before the measurement. Current density is obtained by dividing the total current by the corresponding ECSA. In the HER system, the measurements carried out by the electrochemical workstation were similar to $CO_2RR$, but with slightly different setting parameters. CV would sweep 20 segments from 0 V to −1.0 V with the potential interval of 10 mV. LSV followed the same potential and interval set as CV. The setting parameters of EIS were the same as that of $CO_2RR$. All the data were also converted to RHE. The light source in the photo-electrocatalytic HER system was the same as for $CO_2RR$.

### Large-scale atomistic simulation of hot carrier generation

**Details on the numerical calculation of the hot carrier generation rate.** The large-scale atomistic hot carrier generation simulations were carried out following a recent two-step method developed by Jin et al.[70]:The first step is to calculate the electric field inside the NP and the second step is to use this information to get the hot carrier generation rate using Fermi's Golden Rule (FGR).

For the first step, the NP is considered as a dielectric under an applied external electric field. The nanoparticle is assumed to behave as a dielectric in vacuum with dielectric constant ε (ω) for an electric field of frequency ω. The values of ε (ω) are obtained from experimental data of bulk gold[85]. Within the quasistatic approximation, the electric field inside the nanoparticle can be found by solving Laplace's equation for the electric potential using commercially available software such as COMSOL and imposing the boundary conditions that the electric field at infinity is uniform.

Figure 3g–i shows the absolute value of the electric field in each of the nanoparticle geometries, with the incident electric field pointing in the z direction and frequency tuned to the LSPR. The nanoparticle sizes were chosen such that each contains around 200,000 gold atoms, lying comfortably inside the range of sizes required for the quasistatic approximation to hold. Field enhancement effects are clearly visible at the corners and edges of all nanoparticle geometries.

Once the electric potential has been found, it acts as the perturbation driving the system out of equilibrium. For the second step, the

rate at which electron–hole pairs are generated is found using the FGR, which is evaluated using the Kernel Polynomial Method (KPM)[86] to avoid explicit calculation of eigenfunctions and eigenenergies. The quantum-mechanical properties of the electrons are described by a tight-binding Hamiltonian obtained via a two-center Slater-Koster parametrization of bulk gold[87]. This is a 9-orbital model, taking into account the 5d, 6s, and 6p orbitals, while spin is only taken into account as a degeneracy factor. The atomistic NP is constructed by specifying the bounding facet planes and filling the inside with a face-centered cubic lattice compatible with those atomic planes. This was the approach used to obtain the hot carrier generation rate.

**Fermi's golden rule applied to hot carrier generation.** The electric potential $\Phi$ acts as the external perturbation exciting electrons from initial state i (with energy $\epsilon_i$) to final state f (with energy $\epsilon_f = \epsilon_i + \hbar\omega$). The rate $\Gamma_{if}$ at which these transitions happen is obtained via Fermi's golden rule, using second-order perturbation theory:

$$\Gamma_{if} = \frac{2\pi}{\hbar} |\langle i|\Phi|f\rangle|^2 \delta(\epsilon_i - \epsilon_f + \hbar\omega) \quad (5)$$

The number of hot electrons being generated with energy E is obtained by summing over all possible transitions with final energy E, from occupied states below the Fermi surface to unoccupied states above:

$$N_e(E,\omega) = \sum_{if} \Gamma_{if} f(\epsilon_i)(1 - f(\epsilon_f))\delta(\epsilon_f - E) \quad (6)$$

Every electron created with energy E produces a corresponding hole of energy E- $\hbar\omega$, so the rate of hot hole generation is simply found via $N_h(E, \omega) = N_e(E + \hbar\omega, \omega)$.

**Fermi's Golden rule and KPM.** Knowledge about the eigenstates and eigenenergies requires exact diagonalization of the corresponding Hamiltonian H. The numerical complexity of these algorithms typically limits the number of atoms to a few thousand. In contrast, methods like KPM[86] have been very successful[88] in the last decade at bypassing these limitations by working in real space and using stochastic evaluations of the trace to avoid diagonalization. This offers a very beneficial tradeoff between efficiency and accuracy, and has recently been applied to hot carrier generation[70]. FGR can be expressed in real space as

$$N_e(E,\omega) = \frac{2\pi}{\hbar} \int_{-\infty}^{\infty} d\epsilon \int_{-\infty}^{\infty} d\epsilon' Tr\left[\delta(\epsilon - H)\Phi\delta(\epsilon' - H)\Phi^\dagger\right]\delta(\epsilon' - E)$$
$$\delta(\epsilon' - \epsilon - \hbar\omega) \quad (7)$$

The Dirac delta operators are computed using an expansion in a series of Chebyshev polynomials $T_n$:

$$\delta(\epsilon - H) = \frac{1}{\Delta}\frac{1}{\sqrt{1 - \tilde{\epsilon}^2}}\sum_{n=o}^{M} w_n^M T_n(\tilde{\epsilon}) T_n(\tilde{H}) \quad (8)$$

This kind of expansion requires rescaling of the energy scales to ensure the rescaled Hamiltonian's spectrum lies within the range]−1,1[. The spectrum bounds $E_{max}$ and $E_{min}$ are found beforehand and used to rescale $\epsilon$ and H, denoted by a tilde. w is a weight factor arising from the Kernel Polynomial Method to deal with Gibbs oscillations and guarantee uniform convergence of the series. The trace is calculated with a Stochastic Trace Evaluation (STE). Defining a random vector in the Hilbert space:

$$|\xi\rangle = \frac{1}{\sqrt{N}}\sum_{i=1}^{N} \xi_i |i\rangle \quad (9)$$

such that the random numbers $\xi_i$ are uncorrelated with variance 1 ensures that, on average, the trace of any operator can be evaluated by simply calculating its expectation value with this random vector: Tr (A) = $\langle\xi|A|\xi\rangle$. This process introduces an error bar to the calculation but for large systems it is considerably more efficient than evaluating the trace explicitly.

## Data availability
The authors declare that all data supporting the findings of this study are available in the article and its Supplementary Information. Source data are provided with this paper.

## Code availability
All the tools used for the data analysis are publicly available, and the version and parameters used have been indicated. The software used to obtain the hot carrier generation rates is available on Github: https://github.com/simaomenesesjoao/MaxTB and the specific version and scripts used for this publication can be found on Zenodo: https://doi.org/10.5281/zenodo.10912245. The numerical data produced is available together with the experimental data publicly provided.

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

## Acknowledgements

The authors acknowledge funding and support from the Deutsche Forschungsgemeinschaft (DFG, German Research Foundation) under Germany´s Excellence Strategy—EXC 2089/1–390776260 e-conversion cluster, the Bavarian program Solar Energies Go Hybrid (SolTech), the Center for NanoScience (CeNS), the European Commission through the ERC Starting Grant CATALIGHT (802989), the DAAD German Academic Exchange Center (57573042), the CSC-LMU program and the Alexander von Humboldt foundation. We thank Susanne Ebert and Dr. Lars Allmendinger for the EA and [1]H NMR measurements and Dr. Xiaguang Zhang, Dr. Giulia Tagliabue (EPFL) and Dr. Wenzheng Lu for helpful discussions. Y.K. acknowledges e-conversion (DFG) for the students exchange award program that supported her 3 months research stay at Central South University, China. S.M.J. and J.L. acknowledge funding from the Royal Society through a Royal Society University Research Fellowship URF\R\191004. J.L. acknowledges funding from the EPSRC programme grant EP/W017075/1.

## Author contributions

Y.K. and S.M.J. contributed equally. E.C., J.L. and R.L. supervised the project. Y.K. and R.L. designed and performed the experiments. K.F. and B.N. performed the XRD measurement. M.C. and S.L. conducted the TEM and SEM measurements. Z.L. and K.L. conducted the DFT calculation. Y.K., M.L. and J.F. discussed the data. Y.K. and S.M.J. wrote the paper.

## Funding

## Competing interests

The authors declare no competing interests.
