## [Peer Review File · Nature Communications]

Effect of Crystal Facets in Plasmonic CatalysisREVIEWER COMMENTS

Reviewer #1 (Remarks to the Author):

The manuscript entitled “Crystal Facet Effect in Plasmonic Catalysis”. Before publication following issues need to be addressed.

1. We request authors to perform long term i-t for CO₂RR (without plasmon effect) to verify that whether the AuNPs (all 3 samples) still follow the same order results and, still facet plays important role in increasing FE efficiency/selectivity???
2. Authors claims there is no liquid product formation during CO₂RR (we still curious to know whether any kind of C products are formed during the reaction).
3. After plasmon effect Au OCs shown great enhancement in FE(CO) for CO₂RR, we curious to know whether it is showing same enhancement towards FE(H₂) production after plasmon effect.
4. Authors claims remarkable enhancement was achieved after plasmon effect at low coordinated sites importantly at corners and edges rather than facet. Still need more insight into it.
5. Other than the aforesaid correction, please revise the manuscript for any typo as well as grammatical errors

Reviewer #2 (Remarks to the Author):

Cortes, Lischner and co-workers present a well-designed work on the effects of crystal facets in plasmonic catalysis. The effect of facets has been relatively unexplored, especially when it comes to experiments. Thus, I think this work is novel and impactful.

At the same time, the technicalities of this work are sound. All the angles of investigation seem to have been covered and thus, the article seems suitable for Nat Comms.

One main comment I have: The authors state that in the plasmonic catalysis section (as opposed to electrocatalytic only), the edges and tips tend to play a more dominant role as compared to the facets. While I agree that hot-carrier generation rates at these edges/tips are much higher, the binding of CO₂, COOH, CO intermediates have not been investigated at such sites - the SI presents their reaction energies only on the facets. It would be good to use DFT to check their values at the edges/tips. I'd imagine that the edges/tips formed in the three types of nanocrystals would be different. So, their role would be good to examine to have a complete picture.

Reviewer #3 (Remarks to the Author):

This research article reports the electrocatalytic CO₂ reduction using three different Au nanoparticle

shapes (NCs, RDs, OCs) and studying the light-induced enhancement. Results shows that under dark conditions the main products are CO and H₂ and that the electrocatalytic activity (both current density and FE(CO)) follows the trend RDs > NCs > OCs, in line with the reported activity of the exposed crystal facets (110, 100, 111, respectively). Under light irradiation the highest enhancement was seen for OCs and NCs, while RDs did not show significant deviation from the dark activity. The experimental results are supported by DFT calculations for the molecular mechanism and electromagnetic simulations to reasoning on the optical and hot carrier properties. It is concluded that low-coordinated sites over facets are more important than facets in determining enhancements in plasmonic catalysis. The work is timely and very relevant as tries to systematically addressing how crystal shape and exposed facets influence plasmon-enhanced electrocatalysis. However, some points should be improved (as detailed below) to clarify the reasons behind the light-induced activity/selectivity enhancement observed in this work. The emphasis on low-coordination sites role is not fully convincing as it should be more interconnected with the electric fields enhancements on those sites. Moreover, it should be verified if RDs, showing the lowest light enhancements, have lower edge/facet ratio than NCs or OCs. The role of interband transitions and plasmonic excitation should be also improved, along with a deeper analysis of the temperature effects. A discussion about the influence of sharp angles and hot spots in general affects the hot carrier generation in the considered systems can be added to clarify this point.

1) The origin of the measured current density should be clarified by reporting a CV under Ar bubbling (instead of CO₂) to verify that the current is due to CO₂ electroreduction. The strongest proof would be to run at least one experiment with CO₂ label ¹³C to be sure that the produced CO is coming from CO₂ and not from ligand decomposition.

2) Table S4 reports the “Edge/facet ratio of Au NCs and OCs”. It should be clarified that the ratio is among the area of facet and edges. As the edge/facet ratio has been used to rationalize the light-induced enhancement, it should be also reported for RDs as it appears important to extend the discussion to Au RDs, i.e. the most performing shape under pure electrocatalytic conditions and the one showing the lowest light-induced catalytic enhancement. The number of edges and facets of the 3 shapes are NCs (12e, 6f), RDs (24e,12f), OCs (12e,8f), suggesting that RDs may have a significant edge/facet ratio. If this is confirmed, why then it shows the lowest enhancement? Can be this due to the fact that RDs already started from a high current density and a FE(CO) >90% in the dark? If so, one can compare the current results with new experiments obtained using spherical Au NPs (or similar shape but exposing facets with low CO selectivity) having low dark FE(CO) and see what is the plasmonic enhancement in this case. This would help to clarify the role of shape in electrocatalytic plasmonic enhancement. It would be beneficial to the readers if the authors would explain how area of edge and area of facet were calculated adding a footnote.

3) The experiments using light 405 nm are particularly interesting as highlight a significantly different behavior among the various employed shapes in comparison to the experiments carried out with 525 nm light. These results should be analyzed more carefully as they may contain crucial information. Indeed, the author mention only that the enhancement in Fe was smaller, particularly for Au NCs. “These findings provide further evidence of the remarkable efficiency of plasmon involvement in catalytic

processes and its potential for enhancing catalyst selectivity.” I partially agree with this analysis as looking to Figure S11 and Fig2b one can see that the FE enhancement with 405 nm for RDs and OCs is very similar to those seen with 525 nm, while it seems lower for NCs (supporting an important role of energetic carriers, in agreement also with simulated carrier energy plots-Fig. d-f). This is confirmed also by the Change of partial current density % plots. Why? The authors should argue and better analyze the role and contributions of Au interband transition to the FE enhancement under light irradiation. True that Au OCs experience the higher FE(CO) enhancement under 525 nm (plasmonic excitation), however, the same enhancement is observed with light at 405 nm, i.e. only exciting interband transitions. This can be key to an improved interpretation of the results. What about trying to plot Delta (Change of FE% 525 nm - Change of FE% 405 nm) trying to isolate more plasmonic and interband hot carrier roles? It is also true that ITs are still active at 525 nm, even if at a minor extent. It would be good to report the chronoamperometry measurements used to compute FE and change of current density under 405 nm.

4) The role of local temperature in plasmonic catalysis is critical for assessing the correct reaction mechanism in action under illumination. Here, the authors carried out experiments in dark at various T and compared the selectivity/FE(CO) with results obtained under illumination. They commented that increasing the T, induced a decrease in FE (CO), while under illumination an opposite effect was observed. They argued that this supported the fact that local heating was not the primary factor contributing to the observed FE(CO) enhancement under light conditions. I agree with the authors that this qualitative comparison gives hints about the effect of T on the electrocatalytic activity. However, it also suggests that the role of light-induced hot carriers may be even more important than that considered in the manuscript. Considering the high light intensity employed in the experiments, the local T on the electrode surface might significantly rise from RT. This should be measured to experimentally verify what is the T generated under illumination. As such, a dark thermal experiment at the T measured under illumination should be carried out. This can be important as the increase in T can decrease the solubility of CO₂ and therefore decrease the current density passing through the electrodes and the FE(CO) as shown by the pilot experiments previously discussed. If this is confirmed, it follows that the effect of hot carriers would be even more pronounced than how it is discussed at the moment.

5) The tests for HER are not particularly convincing as the difference in onset potential reported is very low. I see only one measurement. Are the values of onset potential under light/dark statistically solid? As HER involved a different chemistry and mechanism than CO₂ reduction, I believe this data are redundant and not crucial to sustain the present work.

Minor comments

a) Page 12, line 209-211. The sentence “The increase in CO partial current density on Au OCs and NCs resulted from two factors: the enhancement of FE(CO) and the large photocurrent generated by the hot carriers produced upon the LSPR decay.” Is not clear and need to be rephrased. The photocurrent deriving from hot carriers is registered because a chemical reaction happens at the surface of the electrode. Put in this way, the sentence implies somehow that the production of CO and hot carriers are not related. Is not instead the point of the work using hot carrier to facilitate CO₂ reduction and improve CO selectivity?

- b) Caption to Figure S11 is not clear. Are those plot related to experiments using 405 or 525 nm? Could it be that the Y axis values in Figure S11 are missing to be multiplied by 100?
- c) It would be beneficial to the readers if in Figure d-f would be possible to specify what carrier population is from interband transitions and what from intraband ones.
- d) The Title is confusing as does not reflect the findings, i.e. no influence of crystal facets in plasmon-enhanced electrocatalytic CO₂ reduction.

“Crystal Facet Effect in Plasmonic Catalysis”
Nature Communication - Manuscript ID: NCOMMS-23-47021

We would like to thank the reviewers for their thoughtful comments that helped us to improve the quality and clarity of our manuscript. We have performed a series of experiments that provided us helpful results, aiding us to response the reviewers’ questions. Please find in this letter the reviewers’ comments in black, our point-by-point responses in blue and changes to the manuscript/SI in red. We are very happy to see that the three reviewers suggested minor revisions and that all of them found our manuscript suitable for this Journal. We expect that we have fulfilled all the reviewers’ requirements and we expect that our manuscript can be accepted for publication in Nature Communications at this stage.

Reviewer 1

The manuscript entitled “Crystal Facet Effect in Plasmonic Catalysis”. Before publication following issues need to be addressed.

1. Comment: “We request authors to perform long term i-t for CO₂RR (without plasmon effect) to verify that whether the AuNPs (all 3 samples) still follow the same order results and, still facet plays important role in increasing FE efficiency/selectivity???”

Our response:

We conducted long term i-t experiments for CO₂RR under dark conditions using all three Au NPs. Specifically, we selected two potentials with relatively high FE(CO) and current density: -0.67 V_{RHE} and -0.74 V_{RHE}. Each measurement spanned 5 hours, during which, gas products were injected online into the GC every 30 minutes. After the 5h measurement, liquid products were identified using ¹H NMR of the electrolyte. The FE of CO and H₂, as well as the corresponding i-t curves are now presented in Figure S13 (a-f). For all rounds of stability tests, only minimal amounts of HCOO⁻ could be detected by ¹H NMR with correspondingly less than 1% FE. Therefore, only a typical ¹H NMR spectrum of Au OCs is shown here in Figure S13 (g).

Notably, all three types of Au NPs exhibited relatively stable performance over the 5-hour duration of the tests. Under both tested potentials, the Au NPs consistently demonstrated the trend of RD > NC > OC in terms of FE(CO), indicating the important and stable role of facet in increasing the selectivity in electrocatalytic CO₂RR.

Newly added in Results- Electrocatalytic performance of Au NPs: “The long-term performance (i-t curves) for CO₂RR on the three Au NPs was also checked (Figure S13), resulting in the same order for the FE(CO) performance after 5 hours: RD > NC > OC, indicating the high stability of the system.”

Newly added data in the SI; Figure S13:

[Figure S13. Faradaic efficiencies (FE) of CO and H₂ production as well as the corresponding current density on Au NCs (red), RDs (yellow) and OCs (purple) in long term (5h) electrocatalytic CO₂ reduction system under (a,c,e) -0.67 V_{RHE} and (b,d,f) -0.74V_{RHE}. (g) A typical ¹H NMR spectrum of the electrolyte after 5h of experiments. For all rounds, only very limited amount of HCOO⁻ was detected and the corresponding FE were all less than 1%.]

2. Comment: “Authors claims there is no liquid product formation during CO₂RR (we still curious to know whether any kind of C products are formed during the reaction).”

Our response:

Thanks for the comment. We performed long term electrocatalytic CO₂RR and checked for the liquid products with ¹H NMR, as described in the response to the previous comment. Even if there are some liquid products, they are in very limited amounts (with less than 1% FE for 5h reaction). Therefore, we focus more on the FE of CO to discuss the plasmonic effect in the Au NPs catalyzed CO₂RR system. We have now included this information in the new Figure S13 (as shown before).

3. Comment: “After plasmon effect Au OCs shown great enhancement in FE(CO) for CO₂RR, we curious to know whether it is showing same enhancement towards FE(H₂) production after plasmon effect.”

Our response:

Thanks for your comment. Even our primary focus centers on elucidating the variations in FE(CO) - with a relatively lesser emphasis on FE(H₂) - we have included the changes in FE(H₂) in Figure 2(b). The corresponding data is represented through columns featuring diamond grids.

In the specific context of Au OCs and NCs, the columns in Figure 2(b) are positioned below the x-axis, indicating a reduction in FE(H₂). Conversely, for Au RDs, a marginal enhancement in FE(H₂) is observed, albeit considerably less pronounced than the enhancement noted in FE(CO). We acknowledge the significance of this observation and recognize its potential to contribute to a more comprehensive understanding of our findings. As such, we have now included this observation.

Newly added in Results- Plasmon-assisted electrocatalytic performance of Au NPs with different exposed crystal facets: “In parallel, the FE(H₂) notably diminishes on both Au OCs and NCs, as depicted in Figure 2(b). This decline serves as a clear indicator of the suppression of the competitive HER reaction. Conversely, Au RDs show a marginal augmentation in FE(H₂) at elevated potentials; however, it is important to note that this enhancement is significantly less pronounced compared to the concurrent increase observed in FE(CO).”

4. Comment: “Authors claims remarkable enhancement was achieved after plasmon effect at low coordinated sites importantly at corners and edges rather than facet. Still need more insight into it.”

Our response:

Thank you for the helpful comment. To get more insight into this problem, we calculated the Gibbs free energy of CO₂, *COOH and CO on the edges and corners of three Au NPs, as shown

in Figure S10. Interestingly, the activation energy on the edges (0.65 eV) was found to be similar to the value on the Au{110} facet exposed by Au RDs (0.64 eV), both of which were significantly lower than the values on Au OCs {111}(0.74 eV) and NCs {100}(0.68 eV). This finding provides a plausible explanation for the response of Au NPs to plasmon excitation: Given that the energy barrier on the Au {110} facet is comparable to that of the edges, the plasmon-induced hot electrons generated on the edges exhibit a catalytic activity similar to the electrons on the Au {110} facet. Consequently, their influence is limited in improving FE(CO). Conversely, for Au NCs and OCs, plasmon-induced hot electron generation on edges, characterized by lower energy barriers compared to their exposed facets, leads to a notable enhancement in selectivity. Remarkably, OCs with higher edge/facet ratio exhibit more pronounced improvement in FE(CO). The tip site has a high reactivity in CO₂RR with an energy barrier of 0.60 eV for the activation of CO₂, yet the small number of active sites limits its effect in the rate of the CO₂RR. Thus we consider the Au {110} facet to be the key region for electrocatalytic CO₂RR, and the edges more predominantly in plasmon-assisted systems.

Newly added in Results - Study of the mechanism behind plasmon-enhanced electrocatalytic CO₂RR: “In order to understand this change in reactivity, we conducted DFT simulations to study the CO₂RR reactivity of edge and tip sites, as shown in Figure S10. Interestingly, the activation energy on the edges (0.65 eV) was found to be similar to the value on the Au{110} facet exposed by Au RDs (0.64 eV). The similarity in energy barriers between the Au{110} facet and the edges implies that the plasmon-induced hot electrons generated on the edges, have a limited impact on improving FE(CO). Yet the values on Au OCs {111} (0.74 eV) and NCs {100} (0.68 eV) are higher than that on the edges (0.65 eV). Consequently, for Au NCs and OCs, edges with low coordination number – once activated with light - serve as the pivotal juncture where highly active CO₂RR sites coincide with high electric fields and a substantial population of hot electrons. Thus, plasmon excitation can make a difference in the reactivity of these systems by enabling active sites with lower activation barriers than in dark.”

Newly added data in the SI; Figure S10:

[Figure S10. Model of DFT calculation: side views (top half) and top views (bottom half) of 4×2 periodic surface slab including four atomic layers of (a) Au(100) (b) Au (110) (c) Au (111). **Au** cluster model for edge and corner free energy calculation with exposed facets as (d) Au (100), (e) Au (111). (f) Calculated free energy diagrams for the CO₂ reduction process on Au (100), (110) and (111) facets and (g) free energy diagrams for the CO₂ reduction on edges and corners.]

5. Comment: *“Other than the aforesaid correction, please revise the manuscript for any typo as well as grammatical errors.”*

Our response:

Thanks for your kind comment. We revised the typo errors and grammatical errors in the new version of our manuscript.

Reviewer 2

Cortes, Lischner and co-workers present a well-designed work on the effects of crystal facets in plasmonic catalysis. The effect of facets has been relatively unexplored, especially when it comes to experiments. Thus, I think this work is novel and impactful.

At the same time, the technicalities of this work are sound. All the angles of investigation seem to have been covered and thus, the article seems suitable for Nat Comms.

1. Comment: *“The authors state that in the plasmonic catalysis section (as opposed to electrocatalytic only), the edges and tips tend to play a more dominant role as compared to the facets. While I agree that hot-carrier generation rates at these edges/tips are much higher, the binding of CO₂, COOH, CO intermediates have not been investigated at such sites - the SI presents their reaction energies only on the facets. It would be good to use DFT to check their values at the edges/tips. I'd imagine that the edges/tips formed in the three types of nanocrystals would be different. So, their role would be good to examine to have a complete picture.”*

Our response:

Thanks for the reviewer's suggestion. In response to the reviewer's advice, we developed two models of Au nanocluster consisting of Au {100} and Au {111} facets, respectively, as shown in Figure S10 (d-e). Then the CO₂RR reactivity of edge and tip sites was investigated by DFT calculations, the Gibbs energy on edges (or corners) of the two gold nanoclusters do not change much among them. This might be due to the same coordination number on edges of different shapes or exposed facets.

Interestingly, the activation energy on the edges (0.65 eV) was found to be similar to the value on the Au{110} facet exposed by Au RDs (0.64 eV), both of which were significantly lower than the values on the facets of Au OCs {111}(0.74 eV) and NCs {100}(0.68 eV). This finding provides a plausible explanation for the response of our Au NPs upon plasmon excitation. For instance, the similarity in energy barriers between the Au{110} facet and the edges explains both a) the higher FE(CO) in dark for this type of NPs (with the lower activation energy) and b) the limited impact on improving the FE(CO) upon plasmon excitation (as there are many more facet-sites with already low energy in the system). Conversely, for Au NCs and OCs, plasmon-induced hot electron generation on edges - characterized by lower energy barriers compared to their exposed facets - leads to a notable enhancement in selectivity. Remarkably, OCs with higher edge/facet ratios exhibit the most pronounced improvement in FE(CO). Finally, the tip site should have a higher reactivity in CO₂RR due to a free energy change of 0.60 eV for the conversion of CO₂ to *COOH (the lowest among all the considered sites), yet the small number of this type of active sites limits its effect in the rate of the CO₂RR. Thus, our DFT results are in line with the Au {110} facet to be the key region for electrocatalytic CO₂RR and the edges in the plasmon-assisted processes.

Newly added in Results - Study of the mechanism behind plasmon-enhanced electrocatalytic CO₂RR: “In order to understand this change in reactivity, we conducted DFT simulations to study the CO₂RR reactivity of edge and tip sites, as shown in Figure S10. Interestingly, the activation energy on the edges (0.65 eV) was found to be similar to the value on the Au{110} facet exposed by Au RDs (0.64 eV). The similarity in energy barriers between the Au{110} facet and the edges implies that the plasmon-induced hot electrons generated on the edges, have a limited impact on improving FE(CO). Yet the values on Au OCs {111} (0.74 eV) and NCs {100} (0.68 eV) are higher than that on the edges (0.65 eV). Consequently, for Au NCs and OCs, edges with low coordination number – once activated with light - serve as the pivotal juncture where highly active CO₂RR sites coincide with high electric fields and a substantial population of hot electrons. Thus, plasmon excitation can make a difference in the reactivity of these systems by enabling active sites with lower activation barriers than in dark.”

Newly added data in the SI; Figure S10:

[Figure S10. Model of DFT calculation: side views (top half) and top views (bottom half) of 4×2 periodic surface slab including four atomic layers of (a) Au(100) (b) Au (110) (c) Au (111). **Au** cluster model for edge and corner free energy calculation with exposed facets as (d) Au (100), (e) Au (111). (f) Calculated free energy diagrams for the CO₂ reduction process on Au (100), (110) and (111) facets and (g) free energy diagrams for the CO₂ reduction on edges and corners.]

Reviewer 3

This research article reports the electrocatalytic CO₂ reduction using three different Au nanoparticle shapes (NCs, RDs, OCs) and studying the light-induced enhancement. Results shows that under dark conditions the main products are CO and H₂ and that the electrocatalytic activity (both current density and FE(CO)) follows the trend RDs > NCs > OCs, in line with the reported activity of the exposed crystal facets (110, 100, 111, respectively). Under light irradiation the highest enhancement was seen for OCs and NCs, while RDs did not show significant deviation from the dark activity. The experimental results are supported by DFT calculations for the molecular mechanism and electromagnetic simulations to reasoning on the optical and hot carrier properties. It is concluded that low-coordinated sites over facets are more important than facets in determining enhancements in plasmonic catalysis. The work is timely and very relevant as tries to systematically addressing how crystal shape and exposed facets influence plasmon-enhanced electrocatalysis. However, some points should be improved (as detailed below) to clarify the reasons behind the light-induced activity/selectivity enhancement observed in this work. The emphasis on low-coordination sites role is not fully convincing as it should be more interconnected with the electric fields enhancements on those sites. Moreover, it should be verified if RDs, showing the lowest light enhancements, have lower edge/facet ratio than NCs or OCs. The role of interband transitions and plasmonic excitation should be also improved, along with a deeper analysis of the temperature effects. A discussion about the influence of sharp angles and hot spots in general affects the hot carrier generation in the considered systems can be added to clarify this point.

1. Comment: *“The origin of the measured current density should be clarified by reporting a CV under Ar bubbling (instead of CO₂) to verify that the current is due to CO₂ electroreduction. The strongest proof would be to run at least one experiment with CO₂ label ¹³C to be sure that the produced CO is coming from CO₂ and not from ligand decomposition.”*

Our response:

We conducted the electrocatalytic CO₂RR measurement under Ar bubbling, as shown in Figure S14. Across all three types of Au NPs, a noticeable decrease in current was observed, as illustrated in Figure S14 (a-c). Due to the persistence of HER, the current did not diminish to zero. Nevertheless, the results strongly affirm that the observed current on the surface is associated with the bubbling CO₂. Additionally, we investigated the composition of products under Ar bubbling using GC, as shown in Figure S14 (d). The analysis confirmed that more than 98% of the products were identified as H₂. The remaining trace amounts of CO can be attributed to the dissolution of CO₂ in the NaHCO₃ electrolyte before the measurement. These findings provide strong evidence that the generated CO is derived from CO₂ reduction rather than the decomposition of ligands.

Newly added in Results- Electrocatalytic performance of Au NPs: *“Experiments conducted with Ar bubbling further validate that the generated CO originates solely from the reduction of CO₂ rather than from ligand decomposition, as depicted in Figure S14.”*

Newly added data in the SI:

[Figure S14. Control experiment under Ar bubbling. (a-c) Current density under CO₂ and Ar on (a) NCs, (b) RDs and (c) OCs. (d) Corresponding FE of gaseous products.]

2. Comment: “Table S4 reports the “Edge/facet ratio of Au NCs and OCs”. It should be clarified that the ratio is among the area of facet and edges. As the edge/facet ratio has been used to rationalize the light-induced enhancement, it should be also reported for RDs as it appears important to extend the discussion to Au RDs, i.e. the most performing shape under pure electrocatalytic conditions and the one showing the lowest light-induced catalytic enhancement. The number of edges and facets of the 3 shapes are NCs (12e, 6f), RDs (24e,12f), OCs (12e,8f), suggesting that RDs may have a significant edge/facet ratio. If this is confirmed, why then it shows the lowest enhancement? Can be this due to the fact that RDs already started from a high current density and a FE(CO) >90% in the dark? If so, one can compare the current results with new experiments obtained using spherical Au NPs (or similar shape but exposing facets with low CO

selectivity) having low dark FE(CO) and see what is the plasmonic enhancement in this case. This would help to clarify the role of shape in electrocatalytic plasmonic enhancement. It would be beneficial to the readers if the authors would explain how area of edge and area of facet were calculated adding a footnote.”

Our response:

Thank you for your valuable comment. In light of the similar sizes yet different edge lengths of the three Au NPs, we did not calculate the edge/facet ratio as the number of them, instead we measured their edge lengths in the TEM images and calculated the number of exposed atoms on edges and facets, under the assumption of a singular line of atoms per edge. It turns out that RDs also have a high edge/facet ratio. We then conducted DFT calculations on the lower coordination sites - edges and corners of the Au NPs, as shown in Fig. S10. As anticipated by the reviewer, the bare enhancement attributed to plasmon excitation was notably linked to the over 90% FE(CO) on Au RDs in dark conditions: the activation energy on the edges (0.65 eV) is slightly higher than on the Au{110} facet exposed by RD (0.64 eV). However, these 2 values are much lower than the values on the facets of Au OC{111} (0.74 eV) and Au NC{100} (0.68 eV). Therefore, Au RDs showed FE(CO) >90% in dark conditions due to the inherent lower activation barrier of that facet. Indeed, the similarity in energy barriers between the Au{110} facet and the edges implies that the plasmon-induced hot electrons generated on the edges, have a limited impact on improving FE(CO) for these NPs. Conversely, for Au NCs and OCs, plasmon-induced hot electron generation on edges - characterized as sites with lower energy barriers compared to their exposed facets - resulted in a notable enhancement of selectivity. Remarkably, OCs with higher edge/facet ratio exhibited a more pronounced improvement in FE(CO) than NCs, consistent with this scenario.

Newly added in Results - Study of the mechanism behind plasmon-enhanced electrocatalytic CO₂RR: “In order to understand this change in reactivity, we conducted DFT simulations to study the CO₂RR reactivity of edge and tip sites, as show in Figure S10. Interestingly, the activation energy on the edges (0.65 eV) was found to be similar to the value on the Au{110} facet exposed by Au RDs (0.64 eV). The similarity in energy barriers between the Au{110} facet and the edges implies that the plasmon-induced hot electrons generated on the edges, have a limited impact on improving FE(CO). Yet the values on Au OCs {111} (0.74 eV) and NCs {100} (0.68 eV) are higher than that on the edges (0.65 eV). Consequently, for Au NCs and OCs, edges with low coordination number – once activated with light - serve as the pivotal juncture where highly active CO₂RR sites coincide with a high electric fields and a substantial population of hot electrons. Thus, plasmon excitation can make a difference in the reactivity of these systems by enabling active sites with lower activation barriers than in dark.”

Newly added data in the SI:

[Figure S10. Model of DFT calculation: side views (top half) and top views (bottom half) of 4 × 2 periodic surface slab including four atomic layers of (a) Au(100) (b) Au (110) (c) Au (111). **Au** cluster model for edge and corner free energy calculation with exposed facets as (d) Au (100), (e) Au (111). (f) Calculated free energy diagrams for the CO₂ reduction process on Au (100), (110) and (111) facets and (g) free energy diagrams for the CO₂ reduction on edges and corners.]

Table S4. Edge/facet ratio of Au NCs and OCs.

Au NPs on electrodes				NC	OC	RD
Flat expansion view			Edge length a /nm	42.4	39.6	40.0
Area of edge	$24 \times a \times r$	$24 \times a \times r$	$74 \times a \times r$
Area of facet	$6 \times a^2$	$2\sqrt{3} \times a^2$	$8\sqrt{2} \times a^2$
Edge/facet ratio	1.36×10^{-2}	2.52×10^{-2}	2.35×10^{-2}

*Here we assume the edge length is a , the Au atom radius in FCC phase is r , the value of r is 0.144nm. The edge lengths were measured from TEM images.

3. Comment: “The experiments using light 405 nm are particularly interesting as highlight a significantly different behavior among the various employed shapes in comparison to the experiments carried out with 525 nm light. These results should be analyzed more carefully as they may contain crucial information. Indeed, the author mention only that the enhancement in Fe is smaller, particularly for Au NCs. “These findings provide further evidence of the remarkable efficiency of plasmon involvement in catalytic processes and its potential for enhancing catalyst selectivity.” I partially agree with this analysis as looking to Figure S11 and Fig2b one can see that the FE enhancement with 405 nm for RDs and OCs is very similar to those seen with 525 nm, while it seems lower for NCs (supporting an important role of energetic carriers, in agreement also with simulated carrier energy plots-Fig. d-f). This is confirmed also by the Change of partial current density % plots. Why? The authors should argument and better analyze the role and contributions of Au interband transition to the FE enhancement under light irradiation. True that Au OCs experience the higher FE(CO) enhancement under 525 nm (plasmonic excitation), however, the same enhancement is observed with light at 405 nm, i.e. only exciting interband transitions. This can be key to an improved interpretation of the results. What about trying to plot Delta (Change of FE% 525 nm - Change of FE% 405 nm) trying to isolate more plasmonic and interband hot carrier roles? It is also true that ITs are still active at 525 nm, even if at a minor extent. It would be good to report the chronoamperometry measurements used compute FE and change of current density under 405 nm.”

Our response:

Thanks for the insightful comment. We plotted the delta of FE(CO) with 525 nm and 405 nm, as well as the chronoamperometry measurements used to compute FE and change of current

density, as shown in Figure S11. Notably, only NCs show obvious higher enhancement in FE(CO) with 525 nm illumination compared to 405 nm.

To investigate the reason behind this observation, we conducted calculations of energy absorption per atom for each Au NP shape and region as a function of frequency, as shown in Figure S16. The primary distinctions among the three Au NPs primarily emanate from their edges, where absorption undergoes the most significant changes with variations in wavelength. Specifically, nanocubes (NCs) demonstrate the most substantial change, with absorption decreasing from 2906 eV/ns to 860 eV/ns as the wavelength shifts from 540nm to 413nm - a decrease exceeding 2000 eV/ns. In comparison, octahedra (OCs) experienced a decrease of 1140 eV/ns, and rhombic dodecahedra (RDs) decreased by 180 eV/ns. This large variation on NCs may be attributed to the reduced intraband transition, as illustrated in Figure 3(d-f) - the population of intraband transition decreased from 32% at 563 nm to 10% at 496 nm.

Additionally, we simulated the electric potential enhancement under 405 nm illumination. Interestingly, unlike OCs and RDs, the enhancement on NCs led by 405 nm illumination is more pronounced on the facet, characterized by a higher energy barrier, as calculated in response to the previous question. Consequently, under 405nm illumination, fewer hot electrons are generated and they tend to distribute on sites with higher energy barriers. Therefore, compared to 525 nm illumination, fewer CO₂ molecules can be activated, resulting in obviously smaller improvement in FE(CO).

Newly added in Results- Plasmon-assisted electrocatalytic performance of Au NPs with different exposed crystal facets: “To comprehensively investigate the mechanism, the influence of light wavelength on plasmonic catalytic performance was then examined. An LED with a wavelength of 405 nm, operating at the same power as the 525 nm LED (610 mW/cm²), was employed for this purpose. Despite the higher energy of 405 nm photons compared to 525 nm photons, the enhancement observed in FE(CO) was smaller, particularly evident in the case of Au NCs (see Figure S11). To investigate the underlying reasons for this observation, we conducted calculations of energy absorption per atom for each nanoparticle shape and region as a function of frequency, as depicted in Figure S16. The key disparities arise from the edges, where absorption changes significantly with varying wavelengths. Specifically, NCs display the most substantial change, with absorption decreasing by 70% as the wavelength shifts from 540 nm to 413 nm - a decrease exceeding 2000 eV/ns. In comparison, OCs decreased by 1140 eV/ns, and RDs decreased by 180 eV/ns. This large variation in NCs may be attributed to the reduced intraband transition, as illustrated in Figure 3(d-f), where the population decreased from 32% at 563 nm to 10% at 496 nm. Furthermore, simulating electric potential enhancement under 405 nm illumination revealed a more pronounced effect on the facets for NCs (Figure S10). Consequently, fewer hot electrons are generated and tend to distribute on sites with higher energy barriers. Thus, compared to 525 nm illumination, fewer CO₂ molecules are activated, resulting in a significantly smaller improvement in FE(CO). These findings contribute additional evidence underscoring the significance of hot electron generation and high electric field enhancement in our model system.”

Newly added data in the SI:

Figure S11. CO₂RR performance under 405 nm illumination. (a) Change in the absolute value of FE(CO) and FE(H₂) when illuminated by 405 nm LED compared to dark conditions. (b) Percentage change in j(CO) and j(H₂) when illuminated by 405 nm. (c) The difference between FE(CO)_{525nm} and FE(CO)_{405nm}. (d) A typical chronoamperometry measurements under 6 measured potentials used for computing FE and change of current density under 405 nm.

Figure S16. (a-c) Energy absorption per atom for different regions on Au NPs as a function of frequency. (d-f) Electric potential enhancement on Au NPs with 405 nm illumination.

4. Comment: “The role of local temperature in plasmonic catalysis is critical for assessing the correct reaction mechanism in action under illumination. Here, the authors carried out experiments in dark at various T and compared the selectivity/ $FE(CO)$ with results obtained under illumination. They commented that increasing the T , induced a decrease in $FE(CO)$, while under illumination an opposite effect was observed. They argued that this supported the fact that local heating was not the primary factor contributing to the observed $FE(CO)$ enhancement under light conditions. I agree with the authors that this qualitative comparison gives hints about the effect of T on the electrocatalytic activity. However, it also suggests that the role of light-induced hot carriers may be even more important than that considered in the manuscript. Considering the high light intensity employed in the experiments, the local T on the electrode surface might significantly rise from RT. This should be measured to experimentally verify what is the T generated under illumination. As such, a dark thermal experiment at the T measured under illumination should be carried out. This can be important as the increase in T can decrease the solubility of CO_2 and therefore decrease the current density passing through the electrodes and the $FE(CO)$ as shown by the pilot experiments previously discussed. If this is confirmed, it follows that the effect of hot carriers would be even more pronounced than how it is discussed at the moment.”

Our response:

Thank you for this very interesting comment. We fully agree with the reviewer about this point: the effect of the plasmon is underestimated in our manuscript. However,

we would like to reinforce that is very unusual to have a system in which temperature increase does exactly the opposite as the plasmon excitation does. We think this is among the clearest proofs existing in the literature about a non-thermal effect of plasmonic nanoparticles. However, a proper quantification of thermal effects would be challenging under the present experimental conditions. In order to have a better idea of this point, we estimated the temperature increase and measured a new “dark” performance of the system at an even higher temperature (as control).

Ezendam, S. et al. *Advanced Optical Materials*, 2301496, **2023**, proposed that ΔT could be expressed as $\Delta T \propto P\sqrt{N}$, where P represents the power density of illumination, N denotes the numerical density of nanoparticles. In the referenced work, a laser with a power of 0.156 mW/ μm^2 and a numerical density of 450 yield an estimated and measured temperature enhancement of 150 degrees. In our measurement, we have P as 610 mW/ cm^2 and N as 3.6×10^{10} , so the estimated ΔT is approximately 50 degrees. We then conducted the CO₂RR dark measurement at 70°C, where the FE(CO) further decreased to below 20% at all potentials, as shown in the new Fig. S17. For instance, at -0.81V vs RHE the FE(CO) on Au OCs under plasmon excitation is 87% (Fig. 2), while in dark but under a similar surface temperature of around 70°C the FE(CO) is only 5% (new Fig S17). This clearly highlights the massive impact of the plasmon excitation, in line with the reviewer’s comment.

On the other hand, the solubility of CO₂ decreases from 0.704 (mole fraction \times 1000) to 0.185 when the temperature increases from 20°C to 70 °C, according to J. Phys. Chem. Ref. Data 20, 1201–1209 (1991). This is in line with the measured drop in performance at elevated temperatures (new Fig S17) and also supports the pivotal role of hot carriers and electric field in plasmon-assisted catalytic systems as in those systems there is an associated increase in surface temperature that would reduce the solubility of CO₂ and despite of this, the performance improves under illumination.

Newly added in Results- Plasmon-assisted electrocatalytic performance of Au NPs with different exposed crystal facets: “We then estimated the temperature on the surface of the electrode under plasmon excitation to be $\sim 70^\circ\text{C}$ ⁶⁵. At this high temperature, the performance of the CO₂ reduction in dark is even worst, as shown in Figure S17. This could be in part associated to the reduced solubility of CO₂, which drops nearly 4 times when the temperature increases from 20°C to 70°C. As such, the enhancement of FE(CO) under plasmon excitation has even a larger impact than expected, as it needs to counterbalance the negative impact of fewer CO₂ molecules dissolved in solution. To have real dimension, a proper comparison would be for instance, at -0.81V vs RHE the FE(CO) on Au OCs under plasmon excitation is 87% (Figure 2), while in dark but under a similar surface temperature of around 70°C the FE(CO) is only 5% (Figure S17). Hence, while local heating plays a significant role in various LSPR-involved systems, it has the exact opposite trend here reinforcing the non-thermal effects associated in our study.”

Newly added data in the SI; Figure S17:

Figure S17. FE(CO) on Au OCs under room temperature, 30°C, 40°C and 70°C.

5. Comment: “The tests for HER are not particularly convincing as the difference in onset potential reported is very low. I see only one measurement. Are the values of onset potential under light/dark statistically solid? As HER involved a different chemistry and mechanism than CO₂ reduction, I believe this data are redundant and not crucial to sustain the present work.”

Our response:

Thanks for your thoughtful comment. We conducted the HER measurements for 3 times and plotted the HER onset potential for three Au NPs both under dark conditions and 525 nm illumination, as demonstrated in Figure S15. While the specific values of the onset potentials exhibit slight variations in each round of experiments, the consistent observation is a decrease in the onset potential under illumination, with Au OCs and NCs demonstrating more pronounced reductions compared to RDs, supporting the statistical reliability of the results. We totally agree that HER involved a different chemistry and mechanism than CO₂ reduction. But we believe that exploring a different system adds to the universality of our concept - that low-coordinated sites, as opposed to facets, play a more pivotal role in plasmon-assisted catalysis. Therefore, we have included the HER data to offer our readers a more comprehensive understanding.

Newly added in Results- Plasmon-assisted electrocatalytic performance of Au NPs with different exposed crystal facets: “To assess the statistical robustness of our results, we performed each experiment three times. Across each round of experiments, the Au NPs consistently demonstrated the same trend in the change of onset potentials, as depicted in Figure S15.”

Newly added data in the SI:

[Figure S15. Onset potentials for HER on three Au NPs under dark conditions and 525 nm illumination.]

6. Minor comments:

- a) Page 12, line 209-211. The sentence “The increase in CO partial current density on Au OCs and NCs resulted from two factors: the enhancement of FE(CO) and the large photocurrent generated by the hot carriers produced upon the LSPR decay.” Is not clear and need to be rephrased. The photocurrent deriving from hot carriers is registered because a chemical reaction happens at the surface of the electrode. Put in this way, the sentence implies somehow that the production of CO and hot carriers are not related. Is not instead the point of the work using hot carrier to facilitate CO₂ reduction and improve CO selectivity?*
- b) Caption to Figure S11 is not clear. Are those plot related to experiments using 405 or 525 nm? Could it be that the Y axis values in Figure S11 are missing to be multiplied by 100?*
- c) It would be beneficial to the readers if in Figure d-f would be possible to specify what carrier*

population is from interband transitions and what from intraband ones.
d) The Title is confusing as does not reflect the findings, i.e. no influence of crystal facets in plasmon-enhanced electrocatalytic CO₂ reduction.”

Our response:

Thanks for your kind comment.

a) We revised the sentences in the new version of our manuscript to make the information clearer and easier for the readers to understand.

b) We changed the caption to emphasize these are results with 405 nm illumination. The Y-axis has been changed.

c) We have calculated the isolated contributions coming from intra- and interband transitions and added that new information to the text to make it more quantitative. The image has also been updated to clarify which region contributes to the intra- and which to the interband transitions.

d) We thank about the title suggestion. Indeed, we evaluated this possibility before the submission but we decided to keep the one about the role of crystal facets because we addressed this exact discussion in this manuscript: which is the “role” of the facets? No influence of facets could be misleading as for some cases (as RDs) the role of the facet is similar in dark and light. For this reason, we understand that the current title reflects better the discussion we provide in our manuscript.

Newly added in Results - Plasmon-assisted electrocatalytic performance of Au NPs with different exposed crystal facets: “The chopped i-t curves of three Au NPs are presented by Figure 2d, revealing the noticeable increase in current induced by the light.”

Newly added in Results - Study of the mechanism behind plasmon-enhanced electrocatalytic CO₂RR: “For this reason as well, the Au NCs generate the largest portion of intraband transitions (from 10% of total transitions at 496 nm to 32% at 563 nm) compared to the Au RDs (6% to 15%) and the Au OCs (7% to 17%). For the purposes of this counting, electrons are counted as coming from intraband transitions if their energies are above 1.2eV, the specific values of population are demonstrated in Table S5.”

Figure 3 (a-c) Faradaic efficiencies (FE) for CO production at different applied potentials on (a) Au NCs, (b) RDs and (c) OCs at 20°C, 30°C and 40°C. (d-f) Hot electron (solid) and hole (dashed) generation rate for each of the (d) Au NCs, (e) RDs and (f) OCs at different electric field wavelengths. The Fermi energy is set to zero. (g-i) Absolute electric field profile in (g) Au NCs, (h) RDs and (i) OCs at the corresponding LSPR frequencies, in reference to the external applied electric field. All simulated nanoparticles have approximately 200 000 atoms.

Newly added data in the SI:

Figure S11. CO₂RR performance under 405 nm illumination. (a) Change in the absolute value of FE(CO) and FE(H₂) when illuminated by 405 nm LED compared to dark conditions. (b) Percentage change in j(CO) and j(H₂) when illuminated by 405 nm. (c) The difference between FE(CO)_{525nm} and FE(CO)_{405nm}. (d) A typical chronoamperometry measurements under 6 measured potentials used for computing FE and change of current density under 405 nm.

Table S5. Population of hot carriers of intra- and interband transitions.

Au NP sample	Population of hot carriers in intra- and interband transition under various wavelengths							
	563nm		539nm		516nm		496nm	
	inter	intra	inter	intra	inter	intra	inter	intra
Au NCs	68%	32%	80%	20%	85%	15%	90%	10%
Au RDs	85%	15%	85%	15%	91%	9%	94%	6%
Au OCs	83%	17%	85%	15%	89%	11%	93%	7%

REVIEWERS' COMMENTS

Reviewer #1 (Remarks to the Author):

I have no further question for this work.

Reviewer #2 (Remarks to the Author):

The authors have addressed the comments satisfactorily. I'm happy to recommend publication.

Reviewer #3 (Remarks to the Author):

The authors did address all the criticisms from all reviewers and by doing so they greatly improved the clarity and impact of their work.

The manuscript is now suitable for publication in Nat. Comm.